



# Characterisation of dust aerosols from ALADIN and CALIOP measurements

Rui Song[1], Adam Povey[1,2], and Roy G. Grainger[1]

[1]National Center for Earth Observation, Atmospheric, Oceanic and Planetary Physics, University of Oxford, Oxford, OX1 3PU, UK

[2]Now at National Center for Earth Observation, Physics and Astronomy, University of Leicester, Leicester, LE4 5SP, UK

**Correspondence:** Rui Song (rui.song@physics.ox.ac.uk)

**Abstract.** Atmospheric aerosols have a pronounced effect on climate dynamics at both regional and global scales, but the magnitude of these effects is subject to considerable uncertainties. A major contributor to these uncertainties is the incomplete understanding of aerosol's vertical structure, largely due to observational limitations. Spaceborne lidars can directly observe the vertical distribution of aerosols globally, and are increasingly used in atmospheric aerosol remote sensing. As the first

spaceborne High Spectral Resolution Lidar (HSRL), the ALADIN instrument onboard the Aeolus satellite was operational from 2018 to 2023. With its sophisticated design, ALADIN can retrieve aerosol backscatter and extinction coefficients separately without an assumption of the lidar ratio. This study is dedicated to assessing the performance of ALADIN's aerosol retrieval capabilities by comparing them with CALIOP measurements. A statistical analysis of retrievals from both instruments during the June 2020 Saharan dust event indicates good consistency between the observed backscatter and extinction coefficients. A

detailed comparison of extinction coefficients for dust layers reveals that ALADIN is more susceptible to signal attenuation than CALIOP. During this extreme dust event, CALIOP-derived aerosol optical depth (AOD) exhibited large discrepancies with MODIS Aqua measurements. Using collocated ALADIN observations to revise the dust lidar ratio to 63.5 sr, AODs retrieved from CALIOP are increased by 46%, improving the comparison with MODIS data. Further, the combination of measurements from ALADIN and CALIOP can enhance the tracking of aerosols' vertical transport. This study demonstrates the potential for

spaceborne HSRL to retrieve aerosol optical properties. It highlights the benefits of spaceborne HSRL in directly obtaining the lidar ratio, significantly reducing uncertainties in extinction retrievals. This work paves the way for forthcoming spaceborne HSRL missions, particularly the ESA ATLID space lidar (set for a 2024 launch) and Aeolus-2.

## 1 Introduction

Atmospheric aerosols have a pronounced effect on climate dynamics at both regional and global scales. They directly affect

the climate by scattering and absorbing both shortwave and longwave radiation (Ghan et al., 2012; Myhre et al., 2013; Oikawa et al., 2018). Aerosols also have an indirect effect through their interactions with clouds by modifying their microphysical characteristics, radiative properties, and lifetime (Altaratz et al., 2014; Bellouin et al., 2020). Such interactions alter the net radiation fluxes at the top of the atmosphere and the surface. The magnitude of these effects is subject to considerable uncertainties. These uncertainties are attributed to limitations in the description of aerosol properties, the spatio-temporal variation of



aerosols, and particularly, inadequate understanding of the vertical structures of aerosols. The vertical distribution of aerosols is driven by atmospheric transport patterns, residence times, and the efficiency of vertical transport (Koffi et al., 2012) which vary by up to an order of magnitude among models (Textor et al., 2006; Kipling et al., 2016). Minimising the considerable uncertainties in aerosol vertical distributions is crucial for accurately assessing the effects of aerosols on the climate system.

Vertical dispersal patterns of aerosols have become better constrained since the development of lidar technology. Ground-
based lidar networks, such as the European Aerosol Research lidar NETwork (EARLINET) (Pappalardo et al., 2014), the Micro Pulse lidar NETwork (MPLNET) (Welton et al., 2001), and the Asian Dust and Aerosol lidar Observation NETwork (AD-Net) (Sugimoto et al., 2016), provide detailed vertical aerosol profiles on regional scales.

The limitation in spatial coverage of ground lidar was partially overcome with the launch of lidar into orbit. Spaceborne lidars have the advantage of minimal aerosol loading between the instrument and the calibration region. Lidars launched into
orbit include the Lidar In-Space Technology Experiment (LITE) (Winker et al., 1996), the Geoscience Laser Altimeter System (GLAS) (Spinhirne et al., 2005), the Cloud-Aerosol Transport System (CATS) (McGill et al., 2015), and the Advanced Topographic Laser Altimeter System (ATLAS) (Markus et al., 2017). The Cloud-Aerosol LIdar with Orthogonal Polarization (CALIOP) (Winker et al., 2010) instrument onboard the Cloud-Aerosol Lidar and Infrared Pathfinder Satellite Observations (CALIPSO) satellite, launched in 2006, was tailored to offer vertical profile measurements of both clouds and aerosols coin-
cident with other observations in NASA's A-Train. CALIOP emits laser pules toward the Earth's surface, capturing attenuated backscattered signals at 532 and 1064 nm from which the profile of aerosol backscatter and extinction coefficients can be retrieved. CALIOP measures the linear depolarization of the backscattered signals, facilitating the discrimination of cloud phase and identification of non-spherical aerosols (such as mineral dust, volcanic ash, and soot). The Atmospheric LAser Doppler INstrument (ALADIN) (Stoffelen et al., 2005) onboard the European Space Agency's Aeolus mission further advanced this
field by launching a HSRL. Operational from 2018 until 2023, ALADIN was a state-of-the-art Direct Detection Doppler Wind lidar that operated at 355 nm. While its primary focus was detecting wind patterns, this study considers aerosol backscatter and extinction coefficient retrievals from ALADIN and compares them with CALIOP retrievals.

As an elastic backscatter lidar, CALIOP needs the particle extinction-to-backscatter ratio, commonly referred to as the lidar ratio, to accurately interpret the signals. While its value depends on the microphysical characteristics of aerosols, including their refractive index and size distribution, lidar ratio is unaffected by aerosol concentration (Mona et al., 2006). The lidar ratio
enables the derivation of particle extinction coefficients from single-channel backscatter profiles, and is therefore fundamental to accurate estimation of aerosol radiative impact. However, there remain limitations in CALIOP's lidar ratio selection scheme. For example, the use of a single lidar ratio for all dust aerosols introduces bias (Kim et al., 2020) because the lidar ratio is influenced by the mineralogical composition and refractive index of dust particles (Schuster et al., 2012) and particle non-
sphericity (Dubovik et al., 2006). Beyond the limitations associated with selecting a constant lidar ratio for specific aerosol types, CALIOP's extinction retrieval presents additional challenges. There is a minimum AOD detectable by CALIOP, which affects how observations should be compared (Watson-Parris et al., 2018), with the undetected layers having a global mean AOD of $0.031 \pm 0.052$ (Kim et al., 2017).



High Spectral Resolution Lidars (HSRLs) are increasingly recognised for their potential in atmospheric aerosol remote sens-
ing as they separately detect particles and molecules (Shipley et al., 1983; Müller et al., 2014; Wang et al., 2022). A significant
advantage of this technique is that the aerosol retrieval is independent of assumptions regarding the lidar ratio. The aerosol
and cloud retrievals from ALADIN have been systematically validated against a variety of ground-based measurements (Baars
et al., 2021; Paschou et al., 2022; Abril-Gago et al., 2022; Feofilov et al., 2022; Gkikas et al., 2023). The ALADIN instrument
employs a circularly polarized emission, but only detects the co-polar component of the return. Due to this instrument con-
figuration, ALADIN's aerosol retrieval underestimates the aerosol backscatter coefficient for highly depolarized atmospheric
particles (Paschou et al., 2022; Gkikas et al., 2023), including ice crystals, smoke, dust, and volcanic ash. However, this misde-
tection of cross-polar component backscattered signals does not influence the retrieval of the extinction coefficient. The aerosol
processing in ALADIN does not rely on the information of the lidar ratio. Instead, ALADIN is capable of retrieving the lidar
ratio as a variable within its Level-2 aerosol products. However, given that its aerosol retrieval process does not set constraints
on the lidar ratio, the retrieved lidar ratio often exhibits significant fluctuations for a given aerosol layer. One scenario leading
to this variability is when the backscattered signal approaches the instrument's detection threshold. Thus, effective filtering is
essential when analysing ALADIN lidar ratios. Additionally, ALADIN's Level-2 backscatter and extinction coefficients are
subject to independent Quality Control (QC) procedures. Despite these challenges, it has been demonstrated that ALADIN is
capable of retrieving lidar ratios from smoke (Baars et al., 2021), dust (Flament et al., 2021) and marine aerosols (Sun et al.,
75   2023).

This study aims to explore and demonstrate the capabilities of ALADIN in retrieving aerosol optical properties, specifically
the backscatter coefficient, extinction coefficient, and lidar ratio. The CALIOP Level-2 aerosol products, with a 5-km horizontal
resolution, are used as a benchmark. The Saharan dust in June 2020 is chosen as the study area. Firstly, desert dust is the most
predominant aerosol by mass in the atmosphere. Secondly, the lidar ratio of dust exhibits pronounced geographic variations.
Finally, the Saharan dust event of June 2020 serves as a unique challenge, acting much like a stress test for evaluating space
lidar measurements (particularly where the dust layer can fully attenuate the return). In this study, a statistical analysis was
undertaken to compare ALADIN and CALIOP in their retrieval of aerosol backscatter and extinction coefficients. To further
understand the underlying causes of discrepancies in extinction retrievals, a comparison was made between the dust lidar ratio
values assumed by CALIOP and those retrieved by ALADIN. This paper also introduces findings from the combined utilisation
of both spaceborne lidars to trace the vertical transport of a dust plume within a specified region.

This paper is structured as follows. Section 2 introduces the aerosol products of Aeolus-ALADIN and CALIPSO-CALIOP
and analyses the collocation between these two spaceborne lidars. Section 3 highlights the challenges of differentiating between
aerosol and cloud in ALADIN data and proposes a solution by using a dust mask derived from coincident geostationary satellite
observations. Section 4 compares the retrieval of aerosol backscatter and extinction coefficients from ALADIN and CALIOP,
focusing on the Saharan dust event of June 2020. Section 5 provides an in-depth analysis of extinction retrievals at different
altitude layers, utilising collocated measurements from both lidar systems. Section 6 further explores the dust lidar ratio, a
key parameter influencing the observed discrepancies in extinction retrievals, and evaluates these findings by comparing the



AOD with MODIS measurements. Section 7 is dedicated to demonstrating the potential of combining ALADIN and CALIOP measurements in enhancing the tracking of aerosol vertical movement. Finally, Section 8 concludes this paper.

## 2 Data

This section introduces the aerosol products from Aeolus-ALADIN and CALIPSO-CALIOP, followed by a discussion of the collocations between the two instruments.

### 2.1 Aeolus-ALADIN aerosol products

Aeolus was launched into space on 22 August 2018 and concluded its mission on 30 April 2023, operating in a Sun-synchronous orbit at an altitude of 320 km with an inclination angle of 97°. The Aeolus satellite hosted ALADIN as its sole payload, which was equipped with an Nd:YAG laser, emitting narrow-bandwidth UV laser pulses at a wavelength of 355 nm. Completing 16 orbits per day, Aeolus maintained a revisit time of 7 days. The laser was directed at an off-nadir angle of 35° as the primary mission was the sounding of horizontal winds.

Each observation by ALADIN integrates laser shots over a 12-second interval, corresponding to an along-track horizontal resolution of approximately 87 km. Each observation is comprised of 24 vertical bins, with varying vertical resolutions: 0.5 km between 0 and 2 km, 1 km between 2 and 16 km, and 2 km between 16 and 30 km. This spacing was adjustable to meet the requirements of specific scenarios. For instance, the ceiling was increased to 30 km near the Hunga Tonga–Hunga Ha'apai plume (30° S - 0°) in response to the changes observed a few days after the eruption on 15 January 2022 (Legras et al., 2022).

The ALADIN Level-2A products are derived using several algorithms, including the Standard Correction Algorithm (SCA), Standard Correction Algorithm middle bin (SCAmb), and the Maximum-Likelihood Estimation (MLE) (Ehlers et al., 2022). The SCA aerosol retrieval is an algebraic inversion scheme that relies on processing cross-talk-corrected signals from both the Rayleigh and Mie channels (Flament et al., 2021). An assessment over the eastern Mediterranean demonstrated that the SCA backscatter coefficients were in good agreement with ground measurements for horizontally homogeneous, fine spherical particles at altitudes below 4 km. However, the performance of the SCA degrades in the lowermost bins, attributed to either contamination from surface signals or to increased noise levels (Gkikas et al., 2023). Another limitation of the SCA method is that the errors in extinction propagate from the first (uppermost) bin to underlying bins. To address this limitation, the SCAmb method averages extinction values over two consecutive bins. Although this results in a reduction in vertical resolution, the trade-off leads to a significant improvement quality. By adapting the SCA method into a physically constrained optimal estimation framework, the MLE method demonstrates a predominantly positive impact coupled with considerable noise suppression. The enhancements effected by the MLE method largely arise from the imposition of positivity constraints on optical properties and the employment of a bounded lidar ratio (Ehlers et al., 2022).In this work, the Level-2 SCAmb products are used to examine ALADIN's aerosol retrieval performance.



## 2.2 CALIPSO-CALIOP aerosol products

The CALIPSO satellite, with the CALIOP instrument as its primary payload, was launched in 2006 alongside CloudSat,
subsequently joining the A-Train (afternoon constellation). It is approximately 73 seconds behind the MODIS Aqua satellite.
This orbital configuration guarantees frequent collocations between CALIOP and MODIS measurements. The specifics of this
collocation process are detailed in Kim et al. (2017), where the collocated MODIS AOD serves as an additional constraint on
CALIOP extinction retrievals. Due to technical challenges affecting its maneuvering capability, CloudSat exited the A-Train
to a lower orbit in February 2018. By September of the same year, CALIPSO rejoined CloudSat in what is now called as the
C-Train. This orbit is 16.5 km below the A-Train, resulting in a slightly different ground track.

CALIOP Level-2 products include the physical and optical parameters associated with detected aerosol and cloud layers.
The utilisation of the Iterated Boundary Location (SIBYL) algorithm aims to optimise the detection of weakly scattering layers
while maintaining reliable identification of dense layers. Nonetheless, given that SIBYL operates based on a threshold-based
detection mechanism, it may occasionally overlook optically thin features that fall below the detection threshold. Subsequent
to detection, the aerosol layers undergo classification into distinct aerosol types. This classification is identified by the Scene
Classification Algorithm (Kim et al., 2018), a decision-tree based method that takes into account factors such as altitude,
geographical location, surface type, estimated particulate depolarization ratio, and integrated attenuated backscatter. In the
final phase, the Level-2 extinction coefficient is retrieved from the Hybrid Extinction Retrieval Algorithm (HERA) (Winker
et al., 2010; Young et al., 2018). The CALIOP Level 1 data provides a horizontal resolution of 333 m and a variable vertical
resolution: 30 m below 8 km and 60 m in the range of 8 to 20 km. In contrast, the CALIOP Level 2 aerosol products present
a horizontal resolution of 5 km. The vertical resolution is 60 m up to an altitude of 20.2 km and transitions to 180 m between
20.2 km and 30.1 km.

In the CALIOP Level-2 Scene Classification V3 and earlier versions, aerosols are categorised into six distinct categories:
clean marine, dust, polluted continental, clean continental, polluted dust, and smoke (Omar et al., 2009). Each aerosol category
is assigned a specific lidar ratio, along with a corresponding uncertainty value. That scheme tended to misclassify aerosols
in regions with a mixture of different aerosol types (Burton et al., 2012; Nowottnick et al., 2015), and it lacks a mechanism
for identifying stratospheric aerosol types. Such aerosol misclassifications can lead to 30-50% uncertainty in the selected lidar
ratio, introducing bias in CALIOP's retrievals (Rogers et al., 2014; Amiridis et al., 2013; Burton et al., 2013). To address these
shortcomings, the CALIOP V4 Scene Classification Algorithm enhanced aerosol subtyping, expanding the number of aerosol
types to 11, covering both tropospheric and stratospheric aerosols (Kim et al., 2018). V4 also revised the lidar ratios designated
for different aerosol subtypes. Owing to these enhancements, CALIOP V4 demonstrates reduced bias in AOD when compared
to AERONET and Moderate Resolution Imaging Spectroradiometer (MODIS) measurements. In this study, the CALIOP Level-
2 V-4.21 aerosol profiles (*CAL_LID_L2_05kmAPro-Standard-V4-21_V4-21*) are used for comparison against ALADIN aerosol
retrievals.



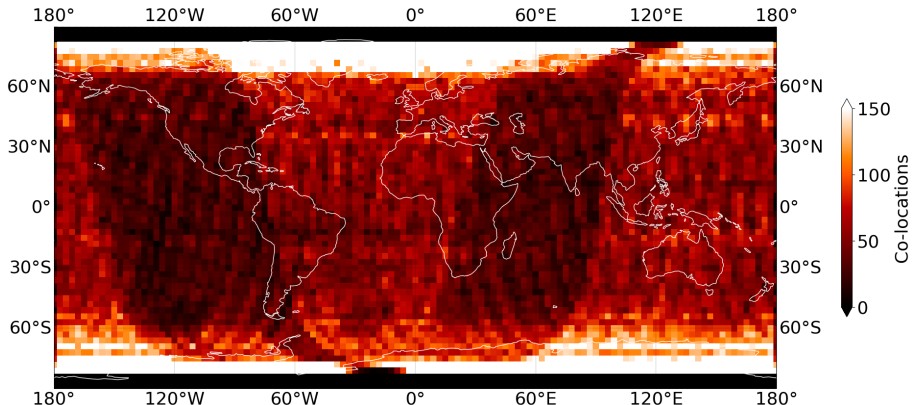

**Figure 1.** Global distribution of collocated ALADIN and CALIOP profiles from 30[th] June 2019 to 28[th] September 2021. The plot, based on a $3° \times 3°$ grid, sets the maximum temporal disparity at 9 hours and the maximum spatial difference at 200 km.

## 2.3 Collocation of Aeolus and CALIPSO

Aeolus performs its overpass of the equator at 06:00 and 18:00 LST, whereas CALIPSO does so at 01:30 and 13:30 LST. The ALADIN lidar has a line-of-sight that is 35° off-nadir towards the Earth's night side. In contrast, the CALIPSO lidar probes the Earth's atmosphere from a nearly nadir angle of 3°. Collocation between Aeolus and CALIPSO represents a balance between the quantity of collocated profiles and their coincidence. In their examination of the scattering ratio profiles from both ALADIN and CALIOP, Feofilov et al. (2022) highlighted the collocation between the two space lidars. They established a collocated database with a spatial distance under 1° and a temporal discrepancy not exceeding 24 hours, based on data between 30[th] June 2019 and 28[th] September 2021. Fig. 1 is a representation of the global distribution of these collocated profiles.

In Fig. 1, it is evident that collocations are concentrated at the poles. The distribution of temporal disparity and spatial distance between collocations, for three latitude bands, are shown in Fig. 2. Between 30° N and 30° S, most collocated observations are within 4 hours and 100 km.

## 3 Aerosol and cloud discrimination

CALIOP's effectiveness in distinguishing between various aerosols and clouds can be largely attributed to its measurements of particle depolarization ratio at 532 nm and its colour ratio between 532 and 1064 nm. With version 4.5 (Tackett et al., 2023), enhancements were made to the CALIOP Level-2 aerosol products, primarily focusing on the improved accuracy of stratospheric aerosol classification.

ALADIN, limited by its single-band observation and its inability to capture particle depolarization information, faces a significant challenge when it comes to discriminating between aerosols and clouds. van Zadelhoff et al. (2023) developed a method known as the ATLID FeatureMask (A-FM) for detecting aerosol and cloud features, intended for use with the forth-



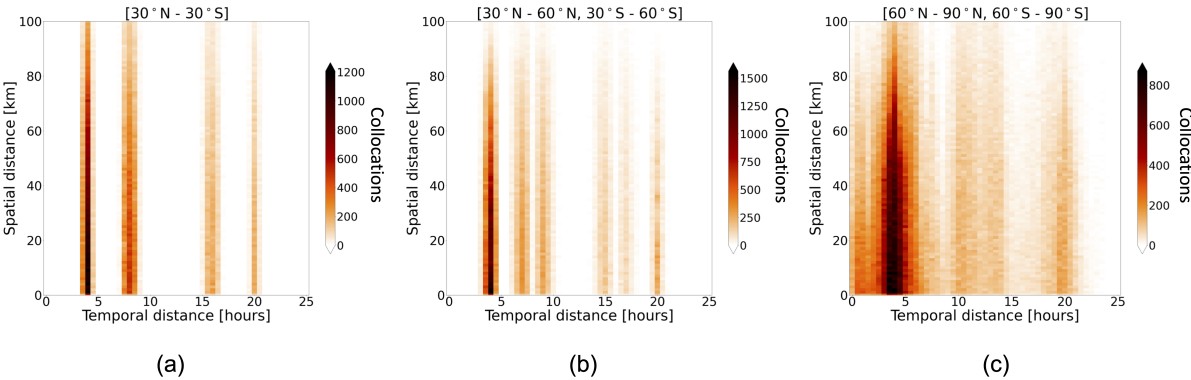

**Figure 2.** Temporal disparity and spatial distance of collocated ALADIN and CALIOP profiles at (a) 30° N - 30° S, (b) 30° N - 60° N and 30° S - 60° S, (c) (b) 60° N - 90° N and 60° S - 90° S.

coming high spectral resolution UV lidar (ATLID) onboard the EarthCARE satellite mission. Initially, the A-FM method was

evaluated using synthetic data from the EarthCARE end-to-end simulator (ECSIM) and real observations from ALADIN's L1 data. It was then adapted into the operational Aeolus FeatureMask (AEL-FM), which is now included in the official L2A Aeolus processor. Another aerosol and cloud discrimination method is proposed in Flament et al. (2021). This method utilises auxiliary meteorological information provided by the European Centre for Medium-Range Weather Forecasts (ECMWF) to identify cloud-free conditions. Both aerosol and cloud discrimination methods highlighted above have undergone updates,

enhancing their accuracy in aerosol and cloud typing. The discrimination methods are planned to be applied during the reprocessing of the ALADIN aerosol products, and both cloud masks will be incorporated into the future releases of ALADIN L2A products.

At the time of this paper's writing, the ALADIN L2A data from the study period does not include the advanced cloud masks described, prompting the exploration of alternatives. In the assessment of Aeolus particle backscatter coefficient retrievals in

the eastern Mediterranean, Gkikas et al. (2023) used the cloud mask product obtained from the Spinning Enhanced Visible and Infrared Imager (SEVIRI) instrument mounted on the Meteosat Second Generation (MSG) geostationary satellite (Schmetz et al., 2002). This cloud mask was used to filter out cloud-contaminated data from ALADIN L2A aerosol products, proving to be an effective approach. This work is focused on the East Atlantic region, which frequently experiences the transport of dense dust plumes from the Sahara. In this context, differentiating between thick dust and clouds using the SEVIRI cloud mask has

proven challenging. As a result, rather than employing a standard cloud mask to filter out cloud-contaminated data for space lidar observations, this study uses a dust mask to identify lidar observations that capture only dust plumes.

Figure 3 provides a comparison of various products used for cloud and dust detection on the 17th June, 2020 at 19:12 UTC. Fig. 3(a) illustrates the SEVIRI dust RGB composite, based on three thermal bands (8.7, 10.8 and 12 μm) from SEVIRI such that shades of pink to violet are interpreted as dust. Fig. 3(b) represents the corresponding SEVIRI cloud mask (CLM)



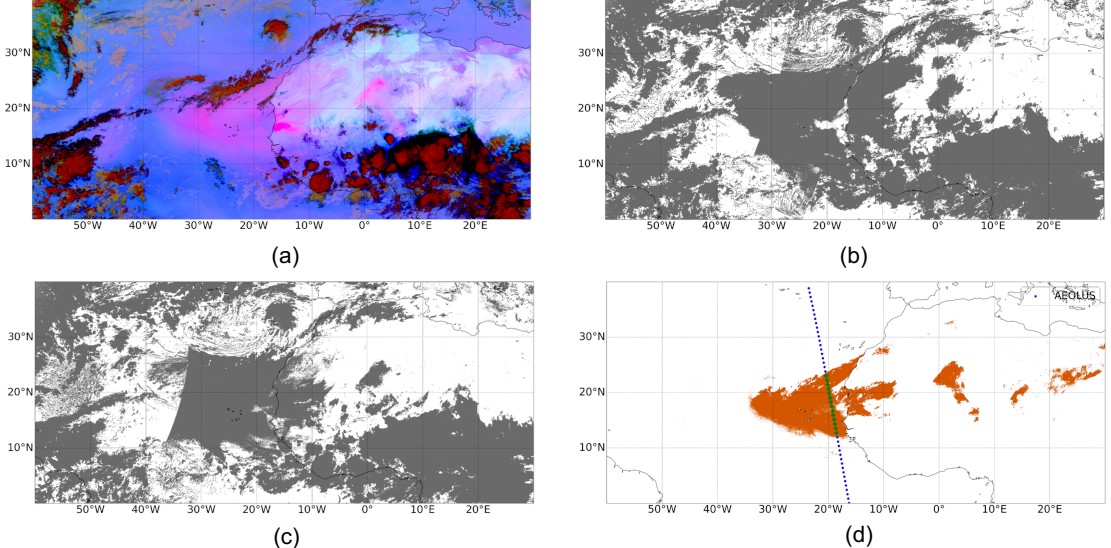

**Figure 3.** Illustration of SEVIRI products and a generated dust flag used on the 17th June 2020 at 19:12 UTC. (a) SEVIRI dust RGB composite, where shades of pink to violet denote dust. (b) SEVIRI CLM cloud mask, highlighting regions identified as clouds in grey. (c) CM SAF cloud mask, showing an alternative cloud identification product by EUMETSAT Satellite Application Facility on Climate Monitoring. (d) Generated dust flag using the method proposed by Ashpole and Washington (2012), illustrating the accurate automatic detection of dust regions over the entire study area. The blue dots in (d) represent the footprint of Aeolus at a horizontal step of approximately 87 km, and the green plus sign marks the location where Aeolus detects dust aerosol in that profile.

product[1], while Fig. 3(c) shows the SEVIRI cloud mask generated by the EUMETSAT Satellite Application Facility on Climate Monitoring (CM SAF)[2]. A comparison of these masks reveals that the CM SAF product has fewer regions misclassified as cloud compared to the CLM product. A significant portion of the dust plume is still incorrectly classified as cloud in both products. Fig. 3(d) displays a dust mask generated using the method of Ashpole and Washington (2012), which can accurately identify dust regions over the entire area automatically. The SEVIRI instrument completes a full-disk scan every 15 minutes,

ensuring a SEVIRI dust flag is available within 7.5 minutes of each ALADIN observation. While ALADIN observations have a horizontal resolution of ∼87 km, the SEVIRI sub-satellite points offer a resolution of ∼3 km. In this study, each geolocation is resampled at a 3 km resolution along the satellite track, and a profile is designated as a dust aerosol observation if 95% of the corresponding resampled footprints are flagged as dust in the relevant SEVIRI data.

---

[1]https://navigator.eumetsat.int/product/EO:EUM:DAT:MSG:CLM

[2]https://navigator.eumetsat.int/product/EO:EUM:CM:MSG:CMA _SEVIRI _V001



## 4   Case study - June 2020 Saharan dust

In June 2020, a large-scale uplift and subsequent transport of dust from the Sahara to the Americas was observed. This event represented the highest AOD for the month of June since 2002. Characterised by continuous emissions over four days, the dust was elevated to altitudes above 6 km due to strong updrafts. The African Easterly Jet facilitated rapid westward long-range transport of the dust (Francis et al., 2020). This study evaluates the accuracy of two space-borne lidar instruments in quantifying this substantial dust event.

Evaluating the accuracy in dust aerosol retrievals between CALIOP and ALADIN is not straightforward. This complexity is largely due to the fact that CALIOP measures the total atmospheric backscattered signals, while ALADIN is designed to only measure the co-polar part of these signals. When non-spherical particles such as dust, volcanic ash, and ice crystals are probed, it can lead to ALADIN underestimating the backscatter coefficients. This was illustrated during the Polly$^{\text{XT}}$ ground lidar experiments conducted in the eastern Mediterranean on the 10$^{\text{th}}$ July 2019, showing ALADIN can underestimate the

aerosol backscatter coefficients by up to 33% when non-spherical mineral particles are recorded (Gkikas et al., 2023).

To address this issue, the method of Abril-Gago et al. (2022) was used to convert between the co-polar part and total particle backscatter coefficient. The formula used to convert between the 355 nm co-polar part and total backscatter coefficient is

$$\beta_{\text{co},355}^{\text{part}} = \frac{\beta_{\text{total},355}^{\text{part}}}{1 + \delta_{\text{circ},355}^{\text{part}}} \tag{1}$$

where $\beta_{\text{co},355}^{\text{part}}$ is the ALADIN 355 nm co-polar part of the particle backscatter coefficient, and $\beta_{\text{total},355}^{\text{part}}$ is the 355 nm total

backscatter coefficient. The circular particle depolarization ratio at 355 nm, $\delta_{\text{circ},355}^{\text{part}}$, is typically not directly measured. It can be estimated if the linear particle depolarization ratio is measured (Mishchenko and Hovenier, 1995), using

$$\delta_{\text{circ},355}^{\text{part}} = \frac{2\delta_{\text{linear},355}^{\text{part}}}{1 - \delta_{\text{linear},355}^{\text{part}}} \tag{2}$$

where $\delta_{\text{linear},355}^{\text{part}}$ is the linear particle depolarization ratio at 355 nm. ALADIN dose not measure the linear particle depolarization ratio, and CALIOP only measures the linear particle depolarization ratio at 532 nm. To address this, a further conversion is

required:

$$\delta_{\text{linear},355}^{\text{part}} = K_\delta \cdot \delta_{\text{linear},532}^{\text{part}} \tag{3}$$

where $K_\delta$ is the spectral conversion factor. Abril-Gago et al. (2022) collected a dataset consisting of measures for $\delta_{\text{linear},355}^{\text{part}}$ and $\delta_{\text{linear},532}^{\text{part}}$ for various aerosol types from the literature, and applied a linear regression to estimate the spectral conversion factor $K_\delta$. For dust, the best linear fit was found to be $K_\delta = 0.82 \pm 0.02$. This value is used in this study for evaluating the

backscatter coefficients obtained from CALIOP and ALADIN.



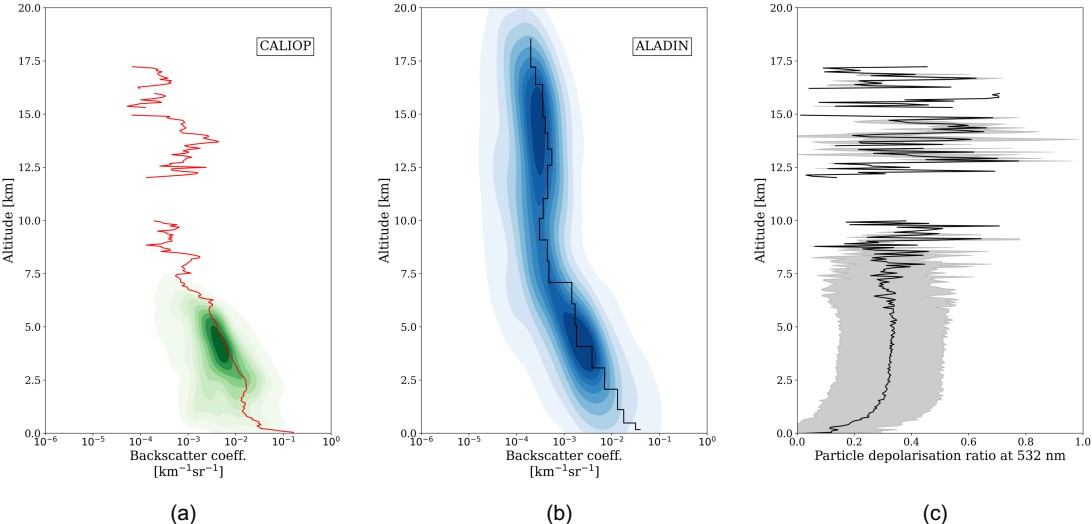

**Figure 4.** Comparison of aerosol backscatter coefficients between CALIOP and ALADIN for the Saharan dust event spanning $14^{th}$ to $24^{th}$ June 2020. The analysis covers the region between $60°$ W and $30°$ E in longitude and $0°$ and $40°$ N in latitude. (a) The green gradient represents the density distribution of particle backscatter coefficients derived from all available CALIOP profiles over the 11-day period, while the red curve indicates the average backscatter coefficient profile at 532 nm. (b) The blue gradient depicts the density distribution of particle backscatter coefficients from all available ALADIN profiles over the same period, with the black curve representing the average backscatter coefficient profile at 355 nm. (c) This panel illustrates the depolarization ratio at 532 nm from CALIOP measurements, where the black curve signifies the mean, and the grey shadow denotes the standard deviation.

Figure 4 illustrates the aerosol backscatter coefficients derived from CALIOP and ALADIN during the 11-day Saharan dust event that began on $14^{th}$ June 2020. The green and blue gradients in each subplot represent the density distribution of dust backscatter coefficients, retrieved at 532 nm for CALIOP and 355 nm for ALADIN, respectively. The respective solid lines depict the mean backscatter coefficients calculated from all retrievals throughout the observed period. For the sake of
comparison, the ALADIN aerosol retrievals in Fig. 4 (a) have been converted from co-polar to total backscatter coefficients, aligning them with the CALIOP aerosol retrievals in Fig. 4 (b). The conversion process involved acquiring $\delta_{linear,532}^{part}$ from the CALIOP measurements depicted in Fig. 4 (c). Between altitudes of 2.5 and 7 km, the depolarization ratio remains fairly constant with a mean value of 0.32. This depolarization ratio aligns with results obtained from other experiments conducted on Saharan dust (Liu et al., 2008), which reported a mean depolarization ratio of 0.32 at the upper part of the dust layer. The
observed decrease in the depolarization ratio in the lower part below 2.5 km is attributed to the mixing of spherical maritime aerosols, known for generally having lower depolarization ratios.



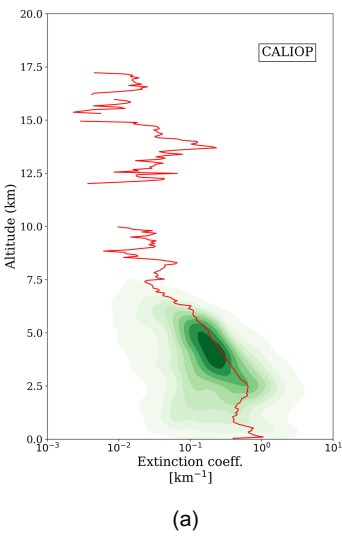
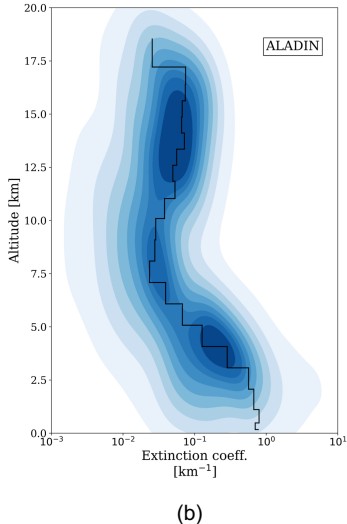

**Figure 5.** Comparison of aerosol backscatter coefficients between CALIOP (a) and ALADIN (b) for the Saharan dust event spanning 14[th] to 24[th] June 2020.

In general, CALIOP and ALADIN show good consistency in detecting dust aerosols, with evidence of dust being uplifted to 7 km. Disparities between CALIOP and ALADIN backscatter coefficients can be primarily traced back to four factors: 1) the spectral difference between 532 and 355 nm; 2) the timing discrepancy as the two instruments are scanning different segments

of the dust plume at different times of the day; 3) ALADIN's coarser sampling rate compared to CALIOP, on both the vertical and horizontal scales, which may cause ALADIN to underestimate aerosol backscatter coefficients at bins with lower aerosol mixing ratios; 4) the conversion from ALADIN's co-polar to total backscatter coefficients involves the use of $K_\delta$, an empirical value of 0.82 obtained from linear fitting for dust aerosols, which could introduce bias during the conversion process. A noteworthy observation from Fig. 4 is the lack of aerosol detection above 8 km by CALIOP, contrasted with ALADIN's ability

to provide an equivalent quantity of aerosol retrievals as the lower atmosphere. This divergence fundamentally originates from the distinct retrieval approaches employed by these two systems. While CALIOP's retrieval relies on an initial aerosol type identification, this constraint is non-existent in ALADIN's retrieval approach. This discrepancy reflects similar issues addressed by Kim et al. (2017), which investigated the bias within CALIOP's column AOD due to undetected aerosol layers. This study focuses on the investigation of aerosol retrievals concentrated within dust layers. Assessing the accuracy of ALADIN's aerosol

retrievals within the upper atmospheric region exceeding the dust layer is beyond the scope of this work. A comprehensive evaluation of whether ALADIN outperforms CALIOP in the detection of weak aerosol signals necessitates an analysis of global aerosol retrievals, including a wide range of aerosol types and distributions. The investigation of this topic will be the subject of future research efforts.



Figure 5 presents a comparison of aerosol extinction coefficients as measured by CALIOP and ALADIN, derived from
the same experimental conditions described in Fig. 4. Although the two instruments generally show a good agreement in
their measurement of extinction coefficients within the dust layer, minor disparities are also apparent. Apart from the spectral
difference, time discrepancy, and contrasting sampling rates, this divergence is largely attributed to the differences inherent in
the extinction retrieval methods of the two instruments. CALIOP's extinction retrieval relies on a predefined lidar ratio tailored
for specific aerosol types. In contrast, ALADIN's backscatter and extinction coefficient retrievals operate independently of
each other. The estimation of the lidar ratio for a given aerosol event can introduce its own set of biases. These biases could be
further magnified in scenarios where the aerosol mixture deviates from the prescribed types. For instance, in this case study,
the lidar ratio in the lower atmosphere below 2.5 km is influenced by both dust and maritime aerosols, leading to an augmented
bias in the lidar ratio estimation.

## 5   Experiments over collocated orbits

Figure 6 displays a pair of collocated orbits, specifically between 50° W and 40° W, on the 24th of June 2020. The overpasses for
Aeolus and CALIPSO are represented in Fig. 6(a) at 20:47 UTC and in Fig. 6(d) at 16:39 UTC, respectively. The background,
as captured by the corresponding SEVIRI dust RGB images, illustrates the relative stability of the dust plume during this
4-hour period. Additionally, this overpass spans a considerable distance across the dust plume that is free of clouds. Fig. 6(b),
(e), (c), and (f) respectively present the extinction coefficients of the dust layer at various altitudes: 2.4 - 3.4 km, 3.4 - 4.4 km,
4.4 -5.4 km, and 5.4 - 6.4 km. These layers have been determined based on the ALADIN grid. Collocated CALIOP retrievals
were upscaled from a resolution of 0.03 km to match this resolution. Layers beneath 2.4 km are not shown due to the reduction
in accuracy from ALADIN resulting from low signal-to-noise ratios. ALADIN and CALIOP extinction retrievals demonstrate
qualitatively good agreement. For Fig. 6(b), both measurements show an extinction of ~0.15 km$^{-1}$, except where ALADIN
observations fail quality-control. This is a common occurrence for the bottom layer of a thick aerosol layer, where signals are
heavily attenuated by the overlying layers. For the middle layers of the dust, ALADIN and CALIOP extinction values display
good agreement in both magnitude and structure. At the top layer between 5.4 - 6.4 km, a very thin dust layer is detected by
both measurements. However, ALADIN exhibits larger values of extinction coefficient, possibly resulting from the temporal
and spatial variability in the measurements. In this instance for the specific lidar overpass, there were no coinciding third-party
aerosol observations available.

Another example of retrieval comparison is illustrated in Fig. 7, featuring descending orbits with CALIPSO overpassing at
04:16 UTC on the 19th of June 2020, and Aeolus overpassing four hours later. This comparison primarily focuses on retrievals
at the peak of this dust event, which is characterised by high AOD values. The extinction retrievals across the upper two
layers (Fig. 7(c,f)), exhibit a consistent level of agreement, reflecting patterns previously observed in Fig. 6. This example also
underscores the divergences in the extinction retrievals from the two instruments within high AOD regions, which become more
pronounced within the middle and bottom layers. In Fig. 7(e), ALADIN retrievals depict a drop within the regions between 14°
N and 20° N. Similarly, for the bottom layer (Fig. 7(b)), ALADIN observations fail to provide quality-controlled retrievals for




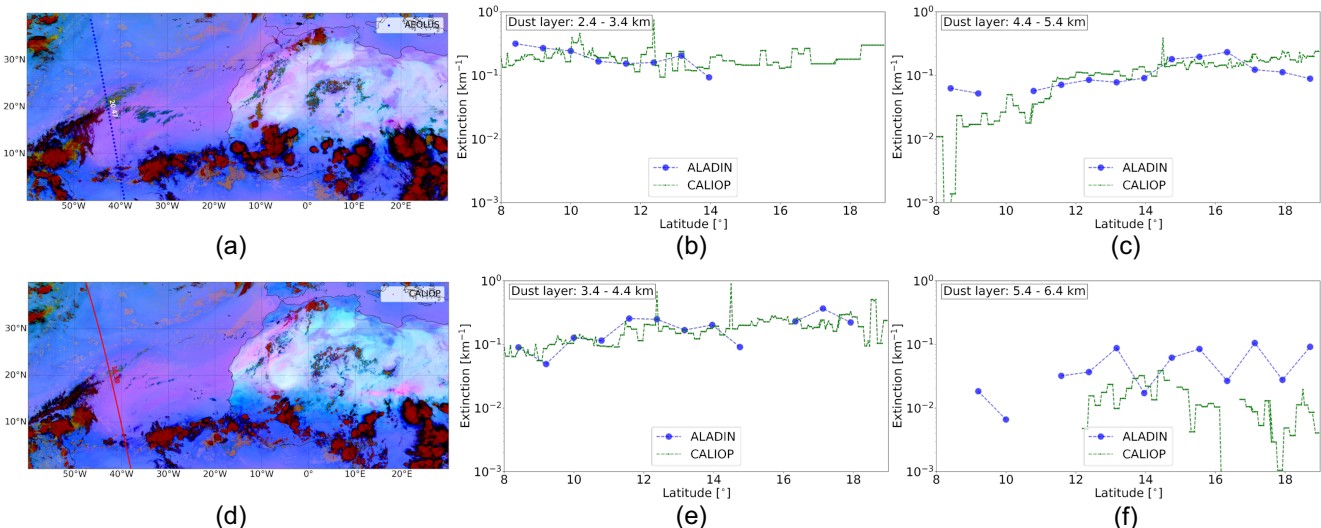

**Figure 6.** Comparison of aerosol extinction retrievals from collocated orbits on the 24[th] June 2020, featuring (a) an Aeolus overpass at 20:47 and (d) a CALIPSO overpass at 16:39, with SEVIRI dust RGB displayed in the background of each. The extinction retrievals from cloud-free regions located between 8° N and 19° N are compared across various altitude layers: (b) 2.4 - 3.4 km, (e) 3.4 - 4.4 km, (c) 4.4 - 5.4 km, and (f) 5.4 - 6.4 km.

an extended area beginning from 10° N and continuing onwards. This example illuminates a common problem with ALADIN extinction retrieval: retrievals at the base of a thick aerosol layer are very likely to be significantly underestimated or excluded by quality control due to low SNRs. A further intriguing insight arises from the layer between 3.4 - 4.4 km (Fig. 7(e)). By

295  filtering out retrievals between 14° N and 20° N, it becomes clear that both instruments efficiently track the spatial evolution of the dust, showing reasonable alignment. This agreement experiences a slight deviation owing to the projection of two datasets with minor geolocation differences onto a linear latitude-based scale. A noteworthy observation is that ALADIN persistently records an extinction coefficient higher by ∼0.2 compared to CALIOP. This discrepancy in absolute extinction coefficients between ALADIN and CALIOP only becomes discernible under two specific conditions: 1) when the extinction within the

300  layer is high - as otherwise the absolute difference substantially decreases, and 2) when the SNR for ALADIN is sufficiently high to surpass the threshold. The hypothesis to explain this phenomenon is that ALADIN, under this given aerosol condition, has higher lidar ratios than CALIOP. A higher lidar ratio inherently leads to elevated extinction coefficients. In light of this, the subsequent section investigate this discrepancy.





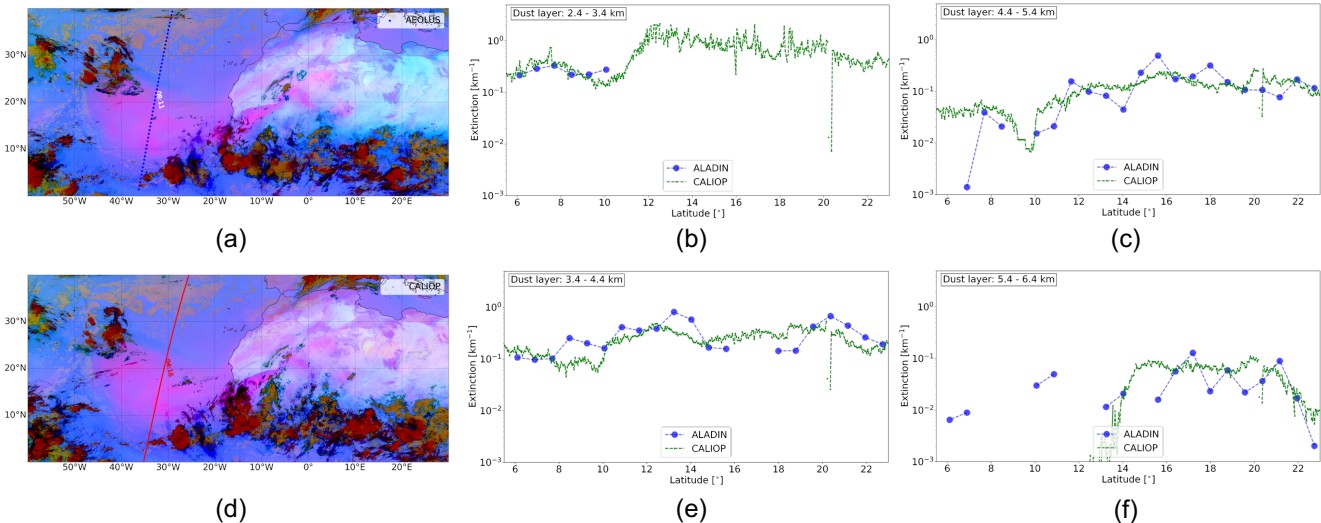

**Figure 7.** Comparison of aerosol extinction retrievals from collocated orbits on the 19$^{th}$ June 2020, featuring (a) an Aeolus overpass at 08:11 and (d) a CALIPSO overpass at 04:16. For details on the background display and altitude layers compared, refer to Fig. 6.

## 6 Lidar ratio and extinction retrievals

305 Figure 8 presents one of the rare instances where both collocated CALIOP profiles and cloud-free MODIS Aqua AOD measurements are available during this dust event. Fig. 8(a) shows the MODIS Aqua AOD at a 3-km resolution, together with the CALIPSO orbit track, while Fig. 8(b) depicts the corresponding CALIOP vertical feature mask. The CALIOP vertical feature mask highlights the dust plume in orange, but it also includes profiles exhibiting fully attenuated bins, represented as black at lower altitudes. To calculate the column AOD from CALIOP extinction retrievals, it is essential to exclude these profiles with 310 fully attenuated bins.

Figure 9 compares AOD between MODIS Aqua and CALIOP for the scene depicted in Fig. 8(a). Each CALIOP profile is paired with the nearest valid, cloud-free MODIS Aqua AOD observations. Within the latitude range of 12° N to 20° N, it is evident that the CALIOP column AOD is considerably underestimated when compared with MODIS Aqua data. Given that CALIOP retrievals have already excluded vertical profiles containing fully attenuated bins, this AOD underestimation cannot 315 be attributed to lost retrievals from the dust's bottom layer.





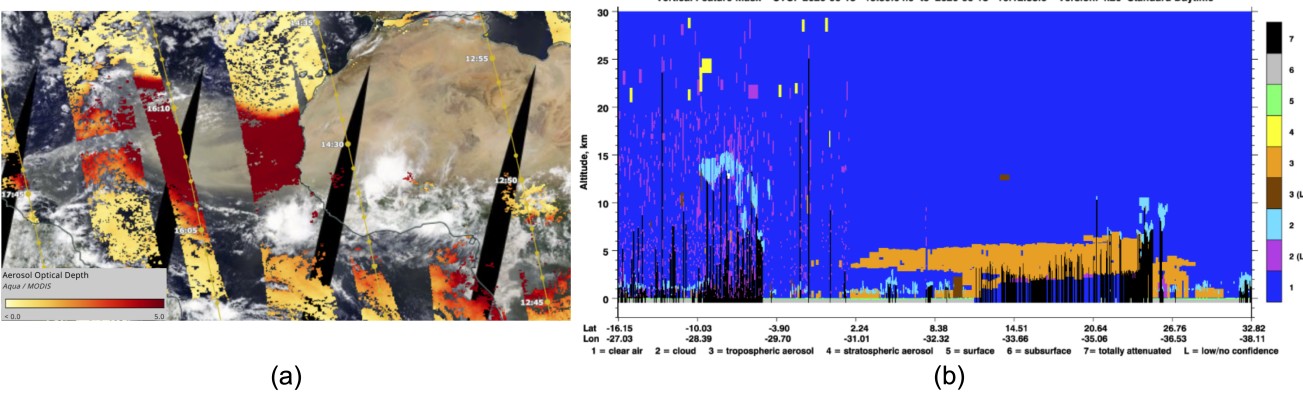

**Figure 8.** Collocated MODIS Aqua and CALIPSO observations at 16:10 UTC on the 18[th] of June 2020. Panel (a) displays the MODIS Aqua cloud-free AOD accompanied by the ascending CALIPSO track (available from NASA Worldview, last accessed on the 3[rd] of July 2023). Panel (b) illustrates the corresponding CALIOP vertical feature mask.

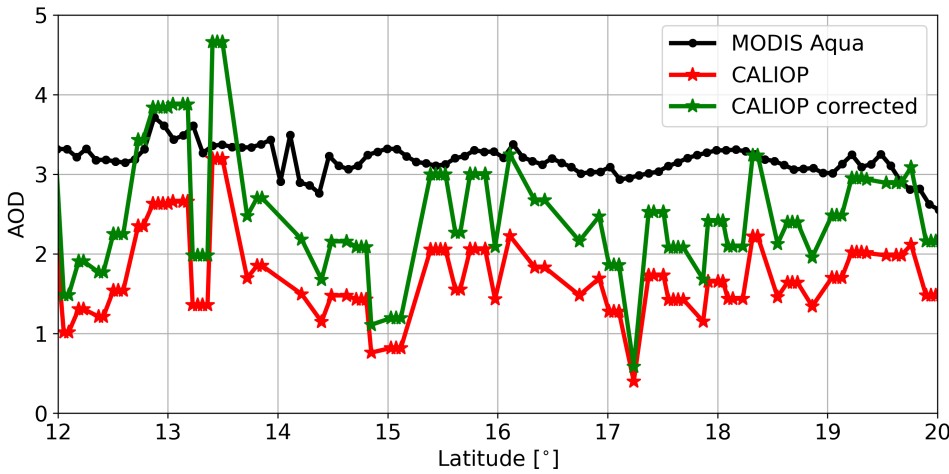

**Figure 9.** Contrast between MODIS Aqua and CALIOP AOD, derived from observational data illustrated in Fig. 8(a). MODIS Aqua AOD is selected exclusively from cloud-free retrievals. The CALIOP AODs have excluded profiles containing fully attenuated bins at any altitudes. The original and corrected CALIOP AODs are shown in red and green, respectively.

In Version 3 and previous releases, a lidar ratio of 40 sr at 532 nm was adopted for CALIOP dust retrievals. Several studies suggest that a larger lidar ratio may be appropriate (Schuster et al., 2012; Papagiannopoulos et al., 2016; Wandinger et al., 2010). With the most recent Version 4 retrieval scheme, CALIOP has increased the lidar ratio of dust to 44 sr for 532 nm (Kim et al., 2018). Dust lidar ratios demonstrate significant regional variability, ranging between 35 and 60 sr (Mamouri et al.,



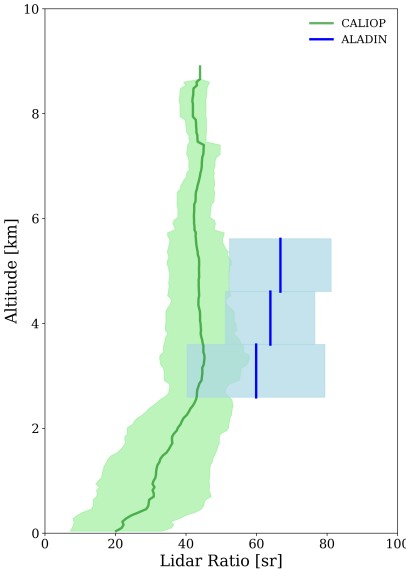

**Figure 10.** Lidar ratios derived for the dust event from 18th - 19th June 2020 with CALIOP depicted in green and ALADIN in blue. The computation of ALADIN lidar ratios incorporated the conversion from co-polar to total backscatter signals.

2013; Nisantzi et al., 2015). Implementing a globally adaptable lidar ratio to accommodate various dust types is complicated, as it requires identifying the source region of the transported dust. Lidar ratios can be extracted from ALADIN observations. However, the derived lidar ratios are frequently noisy and can possess exceptionally small or large values, as the retrieval process is not constrained by the lidar ratios. During the analysis of lidar ratios from ALADIN aerosol retrievals, these noisy values should be filtered out.

Figure 10 presents the lidar ratio calculated between the 18th and 19th of June 2020 for all valid CALIOP and ALADIN retrievals. CALIOP retrievals use an average lidar ratio of 43.5 sr above 2.5 km —- an area less impacted by maritime aerosols and regarded as the dust layer. For ALADIN retrievals, a selective filtering strategy has been implemented, maintaining only data within the 2.4 to 5.8 km altitude range that best characterises the dust layers. Within this particular altitude segment, the mean lidar ratio for dust layers stands at 63.5 sr. Although no established physical equations convert lidar ratios between 355 nm and 532 nm, multiple experiments with ground-based Raman lidars and airborne HSRLs have demonstrated no wavelength dependence of dust lidar ratios at these wavelengths, as detailed in Table 1.

Based on the information supplied in Table 1, it is assumed that $LR_{355\,nm}/LR_{532\,nm} = 1$, thereby justifying the selection of a lidar ratio of 63.5 sr for the correction of CALIOP extinction retrievals at 532 nm. The extinction coefficient $\alpha_{532(corr)}$ is then corrected by multiplying it with $LR_{updated}/LR_{CALIOP}$, where $LR_{updated}$ is set to 63.5 sr and $LR_{CALIOP}$ is derived from CALIOP





| | lidar Ratio (lr) | | |
|---|---|---|---|
| | LR$_{355\,nm}$ | LR$_{532\,nm}$ | LR$_{355\,nm}$ / LR$_{532\,nm}$ |
| Feb 2021 (Haarig et al., 2022) | 47 sr | 50 sr | 0.94 |
| Mar 2021 (Haarig et al., 2022) | 49 sr | 46 sr | 1.07 |
| Jan 2008 (Groß et al., 2011) | 63 sr | 63 sr | 1.0 |
| May 2006 (Tesche et al., 2009) | 55 sr | 56 sr | 0.98 |

**Table 1.** Lidar ratios at 355 and 532 nm derived by various previous studies.

output values. This scaling method is an approximation, as a different lidar ratio can alter the lidar profile and subsequently affect the retrieval process.

Figure 9 displays the revised CALIOP AOD values, represented in green, which are obtained through the correction of extinction retrievals. By applying a correction factor of LR$_{ALADIN}$/LR$_{CALIOP}$, the extinction and AOD values increase by 46%. This augmentation is proportionally applied to both extinction and AOD, thereby measurements exhibiting larger AOD values

witness a more significant increase during the correction, and vice versa. As depicted in Fig. 9, following the correction, a subset of CALIOP AOD values better align with the MODIS AOD. However, there remain certain CALIOP values are significantly lower than the MODIS AOD values.

Fig. 11 shows the vertical distribution of extinction profiles for all CALIOP measurements in Fig. 9 classified into two groups. The first group consists of extinction profiles with a AOD below 1.8, illustrated in Fig. 11 (a), which includes 35

CALIOP profiles. The second group includes extinction profiles with a AOD exceeding 1.8, demonstrated in Fig. 11 (b), containing 24 CALIOP profiles. Both groups capture dust aerosols starting from 1.65 km and dissipating at 5.85 km. The two sets of extinction profiles exhibit a strong similarity in terms of extinction magnitude above 2.4 km. Below 2.4 km, as marked by the red shaded area, the two groups of extinction profiles present considerable discrepancies.

| | Dust layer AOD | |
|---|---|---|
| | layer between 0 and 2.4 km | layer between 2.4 and 7 km |
| Total column AOD < 1.8 | 0.413 ± 0.443 | 1.015 ± 0.365 |
| Total column AOD > 1.8 | 1.094 ± 0.884 | 1.021 ± 0.542 |

**Table 2.** Dust layer AOD for various CALIOP measurements as depicted in Fig. 11.

Table 2 gives the layer AOD values for both groups of CALIOP extinction profiles, those exhibiting higher (> 1.8) and

lower (< 1.8) column AOD measurements. Within the dust layer between 2.4 and 7 km, both groups of measurements present similar layer AODs, 1.021 and 1.015 respectively. Pertaining to the dust layer below 2.4 km, CALIOP measurements begin to reveal the inherent limitation of lidar measurements – the potential for strong attenuation beneath dense aerosol/cloud layers. CALIOP measurements with a column AOD below 1.8 encapsulate those profiles that still feature strongly attenuated bins at the base of the dust layer, despite the implementation of the described filtering strategy. The grouped extinction profile indicate



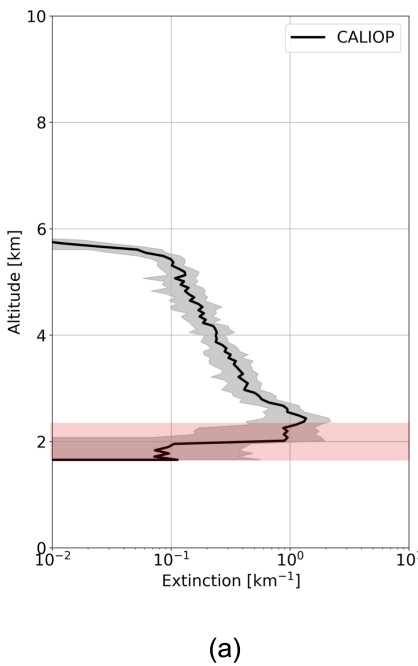

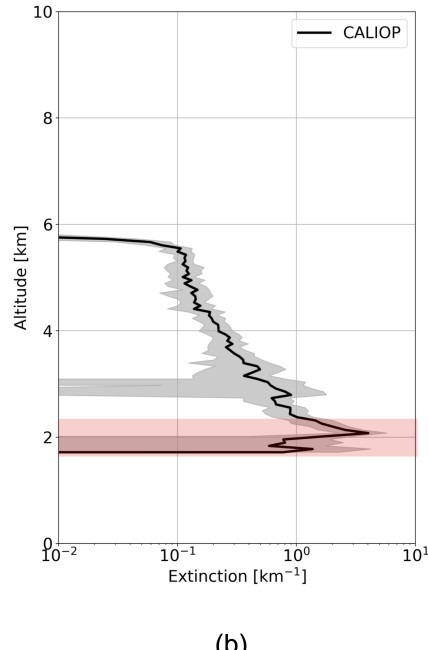

(a)

(b)

**Figure 11.** Averaged CALIOP extinction profiles corresponding to the measurements illustrated in Fig. 9. (a) The average extinction profile for the 35 profiles with a column AOD less than 1.8. (b) The average extinction profile for the 24 profiles with a column AOD exceeding 1.8. Grey shaded areas denotes the standard deviation of extinction. Red highlights areas where notable discrepancies are observed between the two groups of extinction measurements.

a mean layer AOD of 0.413 between the 0 and 2.4 km layer, accompanied by a considerable standard deviation due to the random distribution of strongly attenuated bins along the satellite track. Conversely, the alternative set of measurements devoid of strongly attenuated bins demonstrates a layer AOD of 1.015 between 0 and 2.4 km. These extinction profiles align well with the MODIS column AOD following the correction of extinction values using the ALADIN lidar ratio.

## 7   Vertical transport of dust aerosol

CALIOP, operating as a near-nadir viewing instrument with a narrow cross-track coverage, suffers from limited temporal resolution, with a revisit time of ∼16 days. This limitation constrains CALIOP's capacity to track the localised vertical transport of plumes - such as ash, dust, and smoke - which are frequently linked with extensive horizontal transportation spanning several days to tens of days. Development and preparation for the launch of additional spaceborne atmospheric lidars continues. For instance, EarthCARE is scheduled for launch in 2024, with Aeolus-2 expected to follow near the end of the decade. The

growing presence of atmospheric lidars in space is expected to enhance synergies among different lidars. This would potentially increase the quantity of available observations of aerosol vertical distribution, improving the ability to track the vertical





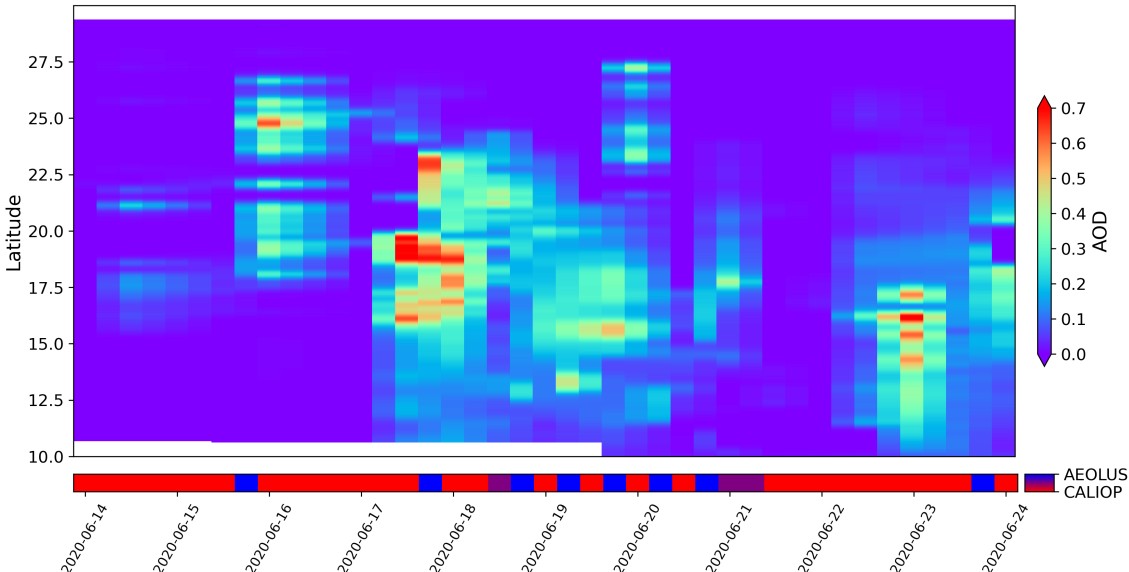

**Figure 12.** Illustration of the synergy between CALIOP and ALADIN layer AOD within the 4.5 - 6.5 km altitude range, between [40°
W, 20° W], covering the 14$^{th}$ to the 20$^{th}$ of June 2020. This vertical layer includes 2 ALADIN bins and 33 CALIOP bins. The lower red-
blue colourbar denotes the contributions from the two distinct lidars, with blue signifying ALADIN, red representing CALIOP, and purple
indicating both. Both measurements have undergone cloud screening to ensure that this figure solely represents the evolution of dust aerosols
within this layer.

transport of aerosols across various locales. Fig. 12 presents a proof-of-concept illustrating the synergy between CALIOP and
ALADIN in tracking the dust plume that penetrated the altitude layer between 4.5 and 6.5 km.

As depicted in Fig. 12, the two satellites align well in detecting the dust aerosols that ascended to a height of 4.5 - 6.5 km on

the 16$^{th}$ and 17$^{th}$ in the area of interest. The peak was noted by the end of the 17$^{th}$, when the layer AOD surpassed 0.7. The dust
aerosols remained confined within this region, and were continuously observed by the two satellites over the subsequent 5 days.
This observation is consistent with the findings in Dai et al. (2022), which used reanalysis data from ECMWF and trajectory
data from HYSPLIT to affirm that the dust plumes were transported within the northeasterly trade-wind zone between latitudes
of 5° N and 30° N and altitudes of 0 and 6 km.

**8   Conclusions**

In 2018, the first spaceborne HSRL ALADIN was launched onboard the Aeolus satellite. This study undertakes an assessment
of ALADIN's performance in retrieving the aerosol backscatter coefficient, extinction coefficient and lidar ratio using its
Level-2 SCAmb products. The aerosol retrievals between ALADIN and CALIOP were compared during the massive Saharan



dust event of June 2020. This is the most intense dust event of the past two decades, lofting dust particles to over 6 km and
transporting dust all the way to the Americas.

The ALADIN does not possess the capability to measure the particle depolarization ratio, constraining its ability to discriminate between aerosols and clouds. This study integrates measurements from the SEVIRI instrument, onboard the MSG geostationary satellite, as a dust feature mask. This operational feature ensures that a SEVIRI dust flag is available for every ALADIN observation, with a maximum temporal discrepancy of 7.5 minutes. This mask allows a more precise evaluation of
ALADIN's observations by isolating data predominantly influenced by dust aerosols despite the low spatial resolution.

ALADIN only detects the co-polar component of backscattered signals, potentially leading to an underestimation of the backscatter coefficient. During the June 2020 Saharan dust case study, the co-polar component of the aerosol backscatter coefficient was converted to represent the total backscatter coefficient. An average taken between 14$^{th}$ - 24$^{th}$ June 2020 reveals a good agreement in backscatter and extinction coefficients from ALADIN and CALIOP, with both instruments showing dust
ascending to 7 km. Discrepancies still persist between the two satellites' retrieval. These discrepancies can be attributed to: 1) The spectral difference, with ALADIN retrieval operating at 355 nm and CALIOP at 532 nm. 2) The different overpass timings of the satellites. 3) The horizontal sampling distance: ALADIN covers 87 km, whereas CALIOP spans 5 km. 4) Uncertainties arising during the conversion from ALADIN's co-polar component to the total backscatter coefficient. When comparing extinction coefficients, an extra contributor to the discrepancy is the lidar ratio. While CALIOP assigned a predefined lidar ratio
for dust, ALADIN's extinction retrieval operates independently of the lidar ratio.

A detailed analysis was conducted to compare the extinction coefficients obtained from collocated ALADIN and CALIOP orbits across various altitude layers. To align with ALADIN's observations, CALIOP's higher vertical resolution data were aggregated into these 1 km layers. Generally, the quality-controlled ALADIN and CALIOP extinction retrievals converge well within the middle and top of the dust layer. However, in the bottom layer ranging from 2.4 to 3.4 km, ALADIN's extinction
retrievals are strongly affected by diminished SNRs.

During this dust event, only one collocated orbit between CALIOP and MODIS was available for a comprehensive AOD comparison. For accuracy, this comparison intentionally omitted CALIOP profiles containing fully attenuated bins from the dust layer's base. Nonetheless, the findings reveal that CALIOP's column AOD is significantly lower than that observed by MODIS Aqua AOD. The lidar ratio is a key parameter in extinction retrievals, with potential to introduce biases that could lead
to disparities in overall AOD calculations. The lidar ratios of dust aerosols were investigated based on observations between 18$^{th}$ and 19$^{th}$ June 2020. CALIOP used a lidar ratio averaging at 43.5 sr. The lidar ratios derived from ALADIN observations showed large variability. Following rigorous filtering, the ALADIN dataset produced a mean lidar ratio of 63.5 sr for the same region and interval.

By applying the ALADIN lidar ratio as a correction for the CALIOP extinction retrievals, the CALIOP-derived AOD re-
trievals increased by 46%, resulting in a closer alignment of a substantial portion of the corrected CALIOP AOD with MODIS AOD. Nonetheless, certain CALIOP profiles continue to reflect AOD values that are significantly lower than those from MODIS. Separating these profiles based on the MODIS AOD revealed that discrepancies in overall AOD values between the two subsets were predominantly sourced from varying extinction retrievals beneath 2.4 km altitude. Given the dense dust



concentration in this layer, CALIOP signals are susceptible to attenuation, leading to potential anomalies in both extinction
and consequent AOD calculations.

This investigation additionally offers a demonstrative application of combining ALADIN and CALIOP observations to derive the vertical transport of aerosols. This methodology serves as a preliminary illustration of the potential collaborative benefits of employing multiple spaceborne lidars to delineate aerosols' spatial trajectories. Such demonstration has significant implications for forthcoming spaceborne HSRL missions, including the ESA EarthCARE's ATLID lidar, set for a 2024 launch,
and the anticipated Aeolus-2 set for deployment by the end of this decade.

*Author contributions.* RS, AP and RG were responsible for conceptualization and methodology. AP and RG supervised this study. RS performed formal analysis and visualization. RS prepared the original draft. RS, AP and RG reviewed and edited the paper. All authors contributed replying to reviewer's comments.

*Competing interests.* The authors declare no competing interests

*Acknowledgements.* This study was funded through NERC's support of the National Centre for Earth Observation, award number NE/R016518/1. CALIOP data obtained from the NASA Langley Research Center Atmospheric Science Data Center.. Aeolus data obtained from ESA Aeolus Online Dissemination Service. This work used JASMIN, the UK collaborative data analysis facility.



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
