# Peer review of "Characterisation of dust aerosols from ALADIN and CALIOP measurements"

_EGUsphere, 2023_

## Author Comment (AC1)

*Response to Referee 2*

The authors present the combination of CALIOP and ALADIN measurements of a dust plume over the Atlantic ocean. They show that ALADIN measurements align well with CALIOP observations as long as proper data filtering as applied. They also re-iterate known limitations of the analysis of CALIOP observations of mineral dust by using ALADIN measurements of the dust lidar ratio to show that the dust lidar ratio in the CALIOP data retrieval is still set too low. The paper would benefit from re-organisation and shortening. Hence, minor revisions are needed.

We greatly appreciate the review and detailed comments provided by the referee. While the limitations of CALIOP in analysing mineral dust are well-established, our study demonstrates the potential of the new ALADIN lidar to address these issues. This aspect of ALADIN, contributing to a more accurate dust lidar ratio from space measurements, represents a novel approach not previously explored in depth. Our responses to the specific comments are as follows.

- I suggest to organise the work in a more conventional way with fewer sections. Please put methods and results into the corresponding sections rather then mixing them up as in the case study. The introduction is also bit lengthy and could be sharpened towards what's relevant for the presented work.

We have restructured the sections of this manuscript. Specifically, the discussion on converting ALADIN co-polar to total backscatter coefficients has been relocated to a new subsection, 2.1 Aeolus ALADIN Aerosol Products. Consequently, Section 2 now includes the following subsections: 2.1 Aeolus ALADIN Aerosol Products; 2.2 CALIPSO CALIOP Aerosol Products; 2.3 Collocation of Aeolus and CALIPSO; and 2.4 Aerosol and Cloud Discrimination. Moreover, the sections comparing ALADIN and CALIOP have been grouped into Section 3, titled 'Case Studies: June 2020 Saharan Dust Event', which includes 3.1 Statistics between ALADIN and CALIOP Retrievals and 3.2 Experiments over Collocated Orbits.

We deleted the last paragraph of the Introduction Section.

- The text includes plenty of figure descriptions that should be covered solely in the figure captions (e.g. lines 193-196, 233-234, 270-275, 306-309, 3043-346). Please omit from the main text.

Following your suggestion, we have carefully reviewed the mentioned sections and omitted repetitive descriptions from the main text.

- Please unify the colour axes in Figs. 1 and 2. The inversion is not very intuitive. Figure 2 might be expanded by a panel that shows all latitudes.

We have unified the colourbars in these figures to avoid any confusion and ensure consistency. However, a panel showing all latitudes was found to be difficult to interpret.

- Why are the authors discussion cloud masks that are not being used for this work? I suggest to stick to what has been used (the MSG-SEVIRI dust mask) and to provide a statement that

other cloud masks have either not yet been available or less useful for your purpose. Figure 3 would need to be revised accordingly.

We appreciate your concern about the discussion of cloud masks in our manuscript. The primary reason for discussing these cloud masks, despite not utilising them in our final analysis, is to justify our methodology. Cloud masks are commonly used to distinguish between cloud and aerosol observations. However, our study deviates from this standard method by employing a specific aerosol type (dust) flag. By presenting the cloud masks and their limitations, we aim to justify and clarify the rationale behind our chosen approach. To clarify which mask is used in our study, the following sentences have been added at the end of subsection Aerosol and cloud discrimination: "In the case studies presented here, the SEVIRI dust mask is used to identify dust-dominated profiles within ALADIN observations. As CALIOP Level-2 APro products already discriminate between aerosol and cloud features, they do not require additional cloud masking".

- Figures 4, 5, and 10 could be improved by adding the number of profiles that contribute at the different height bins. Also, colour scales for the gradients in Figs. 4 and 5 are missing.

We have added colourbars for Fig.4(a) & (b), Fig.5(a) & (b). Profile showing number of valid retrievals at all altitudes have been added to the right margin of each subplot.

- The discussion of Figure 4 - particularly of the particle depolarisation ratio - would benefit from comparisons to findings of SAMUM-2 at Cape Verde.

We have incorporated the findings of SAMUM-2 at Cape Verde from *Ansmann et al., (2011)* into the discussion of the particle depolarisation ratio.

- line 251: shouldn't it be feature type identification?

Yes. We have corrected the sentence.

- I suggest to add lidar curtain plots to Figs. 6 and 7 as those would clearly demonstrate the effect of signal attenuation. It would also be nice if the comparison of extinction coefficients had some quantitative element, such as a correlation plot.

We have now added curtain plots to Fig. 6 and 7. Regarding the comparison of extinction coefficients, we considered incorporating a correlation analysis. A meaningful correlation coefficient requires at least 30 data points. However, we have less than 20 collocated data points available at each altitude layer because of ALADIN's 87 km horizontal resolution. So we removed mention of the correlation coefficients in order to avoid misleading readers. Consequently, we chose a more feasible approach by comparing the extinction coefficients between ALADIN and CALIOP across different altitude layers.

- I don't think that Figure 8 is needed.

We have retained this Fig.8 in our manuscript for two reasons: Firstly, it depicts the horizontal coverage of the dust plume as observed by MODIS, providing a spatial context for our study area. Secondly, the figure includes CALIOP's curtain plots which captured the dust plume. The curtain

plot (b) reveals profiles with a significantly higher number of fully attenuated bins than seen in Fig.6(g) and Fig.7(g), a factor that significantly influences the AOD analysis.

- It is not clear to my what the investigation related to Fig. 11 and Tab. 2 is supposed to tell the readers. Okay, the mean profile shape is different for cases with AOD below or above an arbitrary threshold. But can this be used somehow? If anything, I would expect that type (b) with the higher extinction peak would correspond to the attenuated lower AOD profiles - but it doesn't. This part of the paper left me puzzled and I suggest to omit it.

Fig.11 and Tab.2 were included to address the discussion *"As depicted in Fig. 9, a subset of CALIOP AOD values better align with the MODIS AOD following the correction. However, there remain CALIOP values that are significantly lower than the MODIS AOD values".* Fig.11 and Tab.2 are important for investigating why certain CALIOP AOD values remain significantly lower than those from MODIS even after the lidar ratio correction. Our analysis reveals that for layers between 2.4 and 7 km, both sets of measurements exhibit similar corrected layer-AOD values. However, for the subset with notably lower AOD values, we observed that the layer-AOD between 0 and 2.4 km is substantially lower than the other group. This demonstrates that the lower AOD values are not a result of inadequate lidar ratio correction, but rather due to the inherent limitations of lidar measurements in terms of signal attenuation. Including this analysis in the manuscript underscores the effectiveness of lidar ratio correction in most cases, thereby addressing the initial concern raised in the manuscript.

- I don't think that Section 7 provides any information about the vertical transport of dust aerosol. Fig. 12 is certainly a nice plot that combines the observations of the two platforms. However, it would me more informative if it was to provide information on the longitudinal and height distribution as well. It seems to me that a similar plot could already be produced using much more data from MODIS observations.

We agree that including longitudinal and height distribution in Fig.12 would provide a more comprehensive depiction of dust transport. However, due to the narrow swath and the limited overpasses in the studied area over the 7-day period, it is challenging to expand the plot longitudinally, leading us to limit the longitude between 40° W and 20° W. In terms of height distribution, we faced the issue of low SNR with ALADIN at lower altitudes due to attenuation, and this his can affect the representation of dust transport. Consequently, we focused on the 4.5 to 6.5 km height range to ensure reliability. This plot aims to show that utilising more high-quality lidar profiles in the future can significantly improve our understanding of dust vertical transport. Although our current plot has limitations, it highlights the potential benefits of integrating multiple lidar datasets in such studies.

---

## Author Comment (AC2)

*Response to Referee 3*

The paper "Characterisation of dust aerosols from ALADIN and CALIOP measurements'' aims to assess the performance of ALADIN by comparing with CALIOP, and uses the synergy of both sensors to improve and increase information over an extreme dust transport episode. The manuscript falls within the scope of the journal. However, the presentation and discussion of the paper is not clear and the submitted study is subject to deficiencies. I would recommend publishing considering the major revisions and addressing the specific comments that follow. Furthermore, the text needs rearrangements, especially in terms of the structure in the presentation of the methodology, figures, and discussion in specific sub-sections (see comments below).

We greatly appreciate the review and detailed comments provided by the referee. Our responses to the specific comments are as follows.

Major comments:

- I don't see any direct evidence presented by the authors that supports the statement that "ALADIN is more susceptible to signal attenuation from CALIOP". I suggest removing that statement from the abstract and elsewhere.

Thank you for pointing out the need for further clarification regarding our statement about ALADIN's susceptibility to signal attenuation compared to CALIOP. This statement is derived from our analysis of the extinction coefficient retrievals from both ALADIN and CALIOP, as illustrated in Fig.6 and 7. We noted that ALADIN often has missing retrievals at the base of dust layers (2.4 – 3.4 km layer), primarily flagged for low SNR, indicating signal attenuation is a key factor. In contrast, CALIOP retrievals appear to be less affected by signal attenuation in the studied area. We trust this clarification further substantiates our statement.

- The authors state that their work paves the way for forthcoming spaceborne HSRL missions, particularly the ESA ATLID space lidar (set for a 2024 launch) and Aeolus-2. How does this paper do that?

We deleted this sentence from the abstract. For clarification, our paper showcases the capabilities of Aeolus as the precursor for the next generation of spaceborne HSRLs, like the forthcoming ATLID and Aeolus-2. While Aeolus's primary goal is wind observation, it has introduced the ability to independently retrieve aerosol backscatter and extinction coefficients from orbit, thereby directly measuring the lidar ratio. Despite Aeolus's limitation in spatial resolution, our work demonstrates that its direct lidar ratio measurements can offer improvements over previous technology, as evidenced by our corrected CALIOP extinction retrievals. By establishing the effectiveness of Aeolus's aerosol observation technique, this work paves the way for future spaceborne HSRL missions.

- In abstract and elsewhere, please mention the limitation of ALADIN/Aeolus on retrieving the total particle backscatter coefficient, otherwise the reader will get into confusion (for example, while the term "co-polar backscatter" is not a perfectly valid one, it could be used to distinguish from total backscatter).

The following sentence in the abstract has been rephrased: "…… ALADIN data can be used to estimate aerosol extinction and co-polar backscatter coefficients separately without an assumption of the lidar ratio". Details of ALADIN's limitation on retrieving the total backscatter coefficient has been discussed using Eq (1) – (3)., and this part has been moved to Section 2.1 in the updated manuscript.

- QC flags: The paper extensively utilizes CALIOP and ALADIN, however pre-processing and Quality-Assurance criteria applied on the data used for the comparisons of backscatter and extinction coefficients are not sufficiently presented nor discussed.

We have revised 'Section 2 Data and Methods' to enhance clarity regarding the QA criteria. In the subsection about Aeolus ALADIN data, we have included the following paragraph:

"The quality control of ALADIN's Level-2 SCAmb products involves several criteria: the validity of extinction and backscatter coefficient retrievals; the backscatter-to-extinction ratio (BER); Mie and Rayleigh SNRs; estimated errors in extinction and backscatter coefficients; and the accumulated optical depth. These criteria are comprehensively detailed in (Flamant et al., 2020b). ALADIN's L2A processing strategy has a high sensitivity to errors so that small errors in extinction propagate from one bin to the next, often leading to negative extinction coefficients. To mitigate this issue, an additional filtering step is used in this study to eliminate negative extinction coefficients."

Regarding the subsection of CALIOP data, we have revised the following paragraph:

In this study, the CALIOP Level-2 V-4.21 aerosol profiles APro (CAL_LID_L2_05kmAPro-Standard-V4-21) are used for comparison against ALADIN aerosol retrievals. The Level-2 APro data include a cloud-aerosol discrimination (CAD) score, which we use as a QC flag, selecting only aerosol retrievals with a CAD score less than -20.

- Spectral conversions: Furthermore, assumptions on 532-355 nm spectral dependencies on depolarization, lidar ratio, extinction and backscatter should be supported (preferably using as reference the paper of Floutsi et al., (2023) where averages for these properties using long-term ground-based lidar measurements are given).

Thanks for suggesting this recent reference. Based on the measurements from multiple experiments provided in this reference, we added an extra row of 355 and 532 nm lidar ratio into Table 1.

- For 4 & 5 (aforementioned comments) to be tackled, I would suggest that the authors extensively elaborate on the datasets and methodology sections, to clearly present the processing chains and assumptions leading to the study conclusions.

We have updated Fig.4 and 5 by adding a colourbar for gradient visualisation, changing the ALADIN mean profile colour to red for enhanced visibility, and including a separate plot on the right margins to display the number of valid retrievals at each altitude.

In response to a detailed elaboration on datasets and methodology, the methodological content initially in the first part of Section 4 has been relocated to Section 2. Additionally, we have

restructured 'Section 2 Data and Methods' to enhance clarity regarding the QA criteria. In the subsection of "Aeolus ALADIN aerosol products" and "CALIPSO CALIOP aerosol products", we have included a detailed paragraph about the QA criteria.

- Collocation: Section 2.3 "Collocation of Aeolus and CALIPSO": The paper presents and discusses the following collocation criteria: (1) "3°×3° grid, sets the maximum temporal disparity at 9 hours and the maximum spatial difference at 200 km", (2) "a spatial distance under 1° and a temporal discrepancy not exceeding 24 hours, based on data between 30th June 2019 and 28th September 2021", and (3) "between 30° N and 30° S, most collocated observations are within 4 hours and 100 km". It is not clear at all the selected collocation criteria that are applied in the framework of the study. More important is the authors to discuss atmospheric homogeneity in terms of aerosols and clouds. How is it ensured that the two satellite sensors probe the same air masses? For example, the authors could provide a study on spatiotemporal representativeness in terms of the selected and applied criteria including literature - discussion on NA meteorology. Ensuring that the two systems probe the same air masses is fundamental for the follow-up intercomparison, non assessing it makes the outcome conclusions questionable.

We would like to clarify that the collocation criteria of "a spatial distance under 1° and a temporal discrepancy not exceeding 24 hours" originate from the database developed by Feofilov et al. (2022). While this database originally employed a temporal threshold of up to 24 hours, it allows researchers the flexibility to apply more restrictive temporal criteria as needed. In Fig.1, we have adapted this dataset to a narrower temporal constraint of 9 hours, and regridded the collocation data to a 3°×3° grid. This approach aims to provide a comprehensive global overview of CALIOP and ALADIN collocations, assisting researchers in comparing or integrating data from the two lidars. Specifically, our study employed collocation thresholds of 4 hours and 100 km, focusing primarily on the region between 30° N and 30° S. We have revised Section 2.3 by adding the following sentences for clarification:

"…… Although the dataset utilises a temporal disparity of up to 24 hours, it enables researchers to reduce the temporal threshold. Fig.1 is a representation of the global distribution of these collocated profiles when applying a stricter temporal threshold of 9 hours."

In Fig.1 caption the following sentence has been rephrased:

"……In this plot the dataset is constrained to a temporal disparity of no more than 9 hours, and has been regridded to 3°×3° globally."

We acknowledge that our analysis did not incorporate a trajectory model like HYSPLIT to ensure that both lidars observe the same air parcel. This decision was influenced by the difficulty of identifying the same atmospheric volume with valid retrievals from both lidars, particularly given the specific constraints of the dust event and time period. Our study focuses on comparing the entire vertical profiles of backscatter and extinction retrievals rather than specific layers, which necessitates certain assumptions when comparing profiles from the two lidars. The 4-hour temporal and 100-km spatial disparity criteria used in this study represent the best possible scenario under these conditions, and have been similarly employed in other studies comparing CALIOP and ALADIN, including research by Dai et al. (2022) and Flament et al. (2021).

- Cloud contamination: The dust transport event examined was extreme, however the extensive presence of clouds may affect the scenes examined. Here, with respect to Aeolus Cloud Filtering, three methods are presented and discussed, the (1) SEVIRI CLM cloud mask, (2) the CM SAF cloud mask, and (3) AEL-FM. However, it is not clear which - if not all of the aforementioned cloud-screening datasets are applied. Please elaborate on this aspect, including discussion of the quality of the applied procedures, assumptions, and uncertainties. With respect to CALIOP, which cloud filtering criteria are applied?

We appreciate your attention to the cloud contamination issue and apologise for the confusion regarding our cloud-screening methods. Initially, we examined both the SEVIRI CLM and CM SAF cloud masks but ultimately did not use them in our analysis due to their poor performance in dust areas. Instead, we employed the SEVIRI dust mask for flagging ALADIN observations: a profile is designated as a dust aerosol observation if 95% of the corresponding resampled footprints are flagged as dust in the relevant SEVIRI data. This decision and methodology have been elaborated upon at the end of Section 2.4:

"In the case studies presented here, the SEVIRI dust mask is used to identify dust-dominated profiles within ALADIN observations. As CALIOP Level-2 APro products already discriminate between aerosol and cloud features, they do not require additional cloud masking."

- In the conclusions section, the authors draw generic conclusions on Aeolus and CALIPSO, however their work is based on a single event, which is an extreme one, over a specific domain, and for a time period not exceeding a few weeks. Thus, the outcomes should not be treated as generic since the statistical study lacks the depth to support the argument.

We have revised sentences in the conclusion section to emphasise that our findings and statements are specific to this extreme dust event.

Specific Comments:

- A CALIPSO-based mean depolarization ratio profile is provided, reporting also mean particulate depolarization of 0.32. However, this depolarization is accompanied by a standard deviation ~ ±0.15 which translates to non-pure dust layers apparent in the atmosphere, resulting in particle depolarization values lower than 0.3. How do the authors treat those layers? Treating them as pure-dust layers and applying dust-related conversion factors contaminate the outcomes, so the authors have to address the aerosol mixtures accordingly. Moreover, please mention the pre-processing chains in terms of particle depolarization ratio profiles leading to the non-noisy profile in Fig.4. How do the authors treat larger than 1 and lower than 0 CALIPSO V4 L2 5km depolarization values?

The correction method accounts for aerosol mixing when calculating the conversion factor, and is specifically applied to layers identified as predominantly dust. In Fig.4(c), the standard deviation in depolarization ratios between 2.5 and 7 km altitude mainly arises from observational variations. Below 2.5 km, the decreasing mean depolarization ratio suggests increased mixing of dust with marine aerosols, which is also reflected in the lower lidar ratios at this altitude in Fig.10.

Consequently, the correction method utilises only ALADIN profiles above 2.4 km. This approach is outlined in the manuscript: "...... CALIOP retrievals use an average lidar ratio of 43.5 sr above 2.5 km — an area less impacted by maritime aerosols and regarded as the dust layer. For ALADIN retrievals, a selective filtering strategy has been implemented, maintaining only data within the 2.4 to 5.8 km altitude range that best characterises the dust layers. Within this particular altitude segment, the mean lidar ratio for dust layers stands at 63.5 sr....". The application of CALIOP extinction retrieval corrections is then confined to only dust aerosol layers as seen in Fig.8.

The following sentences have been added in Section 3.1 to elaborate the filtering of depolarization ratio: " ......  After omitting values below 0 and above 1, the depolarization ratio has an average of 0.32 between altitudes of 2.5 and 7 km".

- You can modify Figures 4a, 4b, 5b, and, 5d, to linear instead of logarithmic x-axis scales, add a colorbar for the gradient values, and a second axis reporting on the sample size of profiles resulting in the mean profiles.

We have now included colourbars to represent gradient density values in Figures 4(a), 4(b), 5(a), and 5(b). Additionally, the ALADIN mean profile colour has been changed to red to enhance visibility. The count of valid retrievals contributing to each mean profile has been added to the right margin of each subplot.

In response to your suggestion about the axis scales, we tested with presenting the backscatter coefficients on a linear scale as shown in Fig. S1. However, a linear representation can significantly compromise the visibility of lower backscatter coefficients as they span several orders of magnitude. For this reason, we have decided to retain the logarithmic scale for the figures in our manuscript, as it more effectively displays the full range of data, particularly the comparisons of smaller coefficients. This is consistent with the expectation that particle concentration decreases exponentially with height above the boundary layer.

[Figure]

*Figure S1. CALIOP (a) and ALADIN (b) backscatter coefficients in linear scale.*

- Results presented in Figures 4a, 4b, 5b, and, 5d: you may provide statistical metrics reporting on the intercomparison of backscatter and extinction coefficient profiles (e.g., σ, r2, mean/relative biases, …). Prior to doing this analysis the authors should elaborate how they get CALIPSO to the same horizontal and vertical resolution to Aeolus.

The following statistical analysis has been added to support the discussion of Fig.4:

"...... In general, CALIOP and ALADIN show good consistency in detecting dust aerosols, with evidence of dust being uplifted to 7 km. Within the main aerosol layer from 1.5 to 7.5 km in altitude, the mean backscatter coefficients retrieved by CALIOP and ALADIN show a strong correlation, with an R-square ($R^2$) of 0.967. At ~3.5 km, the altitude with the most valid retrievals, ALADIN's retrieved backscatter coefficient averages 0.004 km−1sr−1. CALIOP, which offers a higher vertical resolution, has an average backscatter coefficient of 0.01 km−1sr−1 when adjusted to match ALADIN's vertical resolution."

The following statistical analysis has been added to support the discussion of Fig.5:

"The two instruments generally show a good agreement in their extinction coefficients within the dust layer, with an $R^2$ value of 0.992 for mean extinction retrievals between 1.5 and 7.5 km altitude. However, some disparities are also apparent. For instance, at the altitude of ~3.5 km, ALADIN has an extinction coefficient of 0.057 km−1 while CALIOP has an extinction coefficient of 0.046 km−1."

- You can apply linear scales also to figures 6(b,c,e,f )and 7(b,c,e,f).

Thank you for your suggestion regarding Figures 6 and 7. As with our tests in Fig.4, we have similar concern in using linear scales for these figures. The extinction coefficients in the case studies of Figures 6 and 7 exhibit significant variations, due to the extensive spatial coverage (over 16 degrees in latitude) and the necessity of maintaining consistent scale ranges across different layers in the subplots. These factors make the logarithmic scale more suitable for effectively representing the data. We hope this explanation clarifies our decision to retain the logarithmic scale in these figures.

- Line 198 and Figure 3: According to the authors the method of Ashpole and Washington (2012) is applied. Since this is a crucial section, please provide discussion on the method, assumption, performance, and uncertainties. Since CALIPSO is used, which is the reason for not implementing CALIPSO aerosol subtype classification as dust identified aerosol layers?

We have expanded the following discussion on the dust flagging method: "This dust flagging method utilises the infrared channels of SEVIRI for the detection of dust events, proving to be effective in consistently identifying moderate to heavy dust outbreaks across the central and western Sahara". The performance and uncertainties of this method are comprehensively detailed in Ashpole and Washington (2012) and we note that this request contradicts that of Reviewer 2 to minimise the discussion of cloud flagging methods.

Regarding CALIPSO's aerosol classification, we refrained from using the dust subtype products due to the known issue of misclassification among various aerosol subtypes (Chen at al., 2010).

- Lines 242-248: This sentence actually is generic to a degree that it doesn't provide any information, since none of the aforementioned source of discrepancies is assessed and no effort in quantifying the effect of each factor is provided in the manuscript. Please elaborate more on this.

The following statistical analysis has been added to support the discussion of Fig.4:
"...... In general, CALIOP and ALADIN show good consistency in detecting dust aerosols, with evidence of dust being uplifted to 7 km. Within the main aerosol layer from 1.5 to 7.5 km in altitude, the mean backscatter coefficients retrieved by CALIOP and ALADIN show a strong correlation, with an R-square (R2) of 0.967. At ~3.5 km, the altitude with the most valid retrievals, ALADIN's retrieved backscatter coefficient averages 0.004 km−1sr−1. CALIOP, which offers a higher vertical resolution, has an average backscatter coefficient of 0.01 km−1sr−1 when adjusted to match ALADIN's vertical resolution."

- Please describe clearly in Section 6 how you apply corrections to CALIPSO and provide the formulas.

We have updated the relevant paragraph to articulate the correction process more clearly:
"The extinction coefficient $\alpha_{532(corr)}$ is then corrected by multiplying it with $LR_{updated}/LR_{CALIOP}$, where $LR_{updated}$ is set to 63.5 sr and $LR_{CALIOP}$ is derived from each individual CALIOP profile. This scaling method is an approximation, as varying the lidar ratio can influence the lidar profile by impacting the backscatter retrieval during the Klett inversion process. This alteration in backscatter retrieval, in turn, affects the subsequent extinction retrieval.

- Please take care of the units in the manuscript, in some places they are missing.

We have checked the units throughout the manuscript.

---

## Author Comment (AC3)

**Response to Referee 4**

The paper of Song et al exploits the aerosol spin-off products for the European wind lidar mission Aeolus for a specific extreme event, namely a heavy Sharan dust outbreak observed over the Atlantic. To compensate the drawbacks of the wind lidar Aladin in the vicinity of non-spherical particles, the authors use the polarization observations from NASA CALIPSO mission to correct Aeolus' co-polar backscatter coefficient and SEVIRI dust mask as a cloud screening proxy.

The paper is of interest for the scientific community, exploits the synergy between different space-born profiles and describes an intense extreme event in a changing climate based on vertically resolved optical properties. It furthermore shows, how different sensors could be used in a synergistic way to retrieve optimized aerosol profiles. It is this worth publishing, however, only after addressing the issues listed below.

We greatly appreciate the review and detailed comments provided by the referee. Our responses to the specific comments are as follows.

Major/General comments:

- Most of the comparisons, especially of extinction coefficient are plotted on logarithmic scale in separated plots. However, by doing so, it is not possible to see major differences in case of strong backscatter and extinction as it is the case for in this paper. Thus, comparisons should be shown in linear scale an, maybe divided by low and high values, to allow the reader to see, how well the results agree. Best, also in the same Figure. Later you state, that „Assessing the accuracy of ALADIN's aerosol retrievals within the upper atmospheric region exceeding the dust layer is beyond the scope of this work." Thus, there is no need to use a log scale.
  At least, I cannot follow many conclusions you have drawn based on the log-based figures you provided.

Thank you for your comments regarding the linear/logarithmic scale of plots (Fig.4 – Fig.7) in our manuscript. We tested with presenting the backscatter coefficients (Fig.4 in the manuscript) on a linear scale as shown below in Fig. S1. However, a linear representation can significantly compromise the visibility of lower backscatter coefficients as they span several orders of magnitude. For this reason, we have decided to retain the logarithmic scale for the figures in our manuscript, as it more effectively displays the full range of data, particularly the comparisons of smaller coefficients. This is consistent with the expectation that particle concentration decreases exponentially with height above the boundary layer.

[Figure]

*Figure S1. CALIOP (a) and ALADIN (b) backscatter coefficients in linear scale.*

To support the conclusions illustrated in the figures, we added quantitative analyses into our discussion. Following Fig. 4 the following sentences have been added:

"...... In general, CALIOP and ALADIN show good consistency in detecting dust aerosols, with evidence of dust being uplifted to 7 km. Within the main aerosol layer from 1.5 to 7.5 km in altitude, the mean backscatter coefficients retrieved by CALIOP and ALADIN show a strong correlation, with an R-square (R2) of 0.967. At ~3.5 km, the altitude with the most valid retrievals, ALADIN's retrieved backscatter coefficient averages 0.004 km−1sr−1. CALIOP, which offers a higher vertical resolution, has an average backscatter coefficient of 0.01 km−1sr−1 when adjusted to match ALADIN's vertical resolution."

Following Fig. 5 the following sentences have been added:

"The two instruments generally show a good agreement in their extinction coefficients within the dust layer, with an R2 value of 0.992 for mean extinction retrievals between 1.5 and 7.5 km altitude. However, some disparities are also apparent. For instance, at the altitude of ~3.5 km, ALADIN has an extinction coefficient of 0.057 km−1 while CALIOP has an extinction coefficient of 0.046 km−1."

- In my opinion, the first part of the section 4, case study is a methodological part and should be put in a respective section. This section should be expanded with respect to CALIPSO observation which have been used: E.g., the quality controls are not clearly described. I can't figure out which CALIOP cloud screening is applied.

We have made several revisions to 'Section 2 Data and Methods'. The methodological content initially in the first part of Section 4 has been relocated to Section 2. This content is now appropriately positioned at the end of the subsection titled "Aeolus ALADIN Aerosol Products".

Additionally, we have revised 'Section 2 Data and Methods' to enhance clarity regarding the QA criteria. In the subsection about Aeolus ALADIN data, we have included the following paragraph:

"The quality control of ALADIN's Level-2 SCAmb products involves several criteria: the validity of extinction and backscatter coefficient retrievals; the backscatter-to-extinction ratio (BER); Mie and Rayleigh SNRs; estimated errors in extinction and backscatter coefficients; and the accumulated optical depth. These criteria are comprehensively detailed in (Flamant et al., 2020b). ALADIN's L2A processing strategy has a high sensitivity to errors so that small errors in extinction propagate from one bin to the next, often leading to negative extinction coefficients. To mitigate this issue, an additional filtering step is used in this study to eliminate negative extinction coefficients."

Regarding the subsection of CALIOP data, we have revised the following paragraph:

In this study, the CALIOP Level-2 V-4.21 aerosol profiles APro (CAL_LID_L2_05kmAPro-Standard-V4-21) are used for comparison against ALADIN aerosol retrievals. The Level-2 APro data include a cloud-aerosol discrimination (CAD) score, which we use as QC flags, selecting only aerosol retrievals with a CAD score less than -20.

- Furthermore, have you used mean Calipso depol profiles for correction or did you make a case by case correction? It is not clearly stated.

Thank you for highlighting this ambiguity in Section 6 regarding the lidar ratio correction. To clarify, we have applied the CALIOP extinction coefficient correction on a case-by-case basis, using depolarization ratios from each individual profile rather than an averaged mean value. To make this clear in the manuscript, we have revised the sentence to explicitly state: "…… where $LR_{updated}$ is set to 63.5 sr and $LR_{CALIOP}$ is derived from each individual CALIOP profile."

- In Section 6, important information is missing, e.g. on how the columnar AOD is calculated from Calipso profiles which obviously are not available down to the ground. Currently, the section is really misleading.

The CALIPSO column AOD is derived by integrating the 532 nm aerosol extinction profile from the 5 km Aerosol Profile Products. Importantly, we have excluded profiles containing fully attenuated bins. This detail has now been added to the last sentence in the first paragraph of Section 4 (updated Section number).

- As the authors focus on a specific atmospheric scene at a specific time of the Aeolus mission, conclusions drawn should not be too general.

We have revised sentences in the conclusion section to emphasise that our findings and statements are specific to this extreme dust event.

Specific comments:

- Please invert color scale, in all other plots of this color map high values are dark and low ones light.

We have now inverted the colourbar in Fig.1 to avoid any confusion and ensure consistency.

- Fig. 4b: Please use a different color for the mean, hardly seen. And please use contours instead of gradients as wording.

We have revised Fig. 4(b) & Fig.5(b) by changing the colour of the mean profiles to red and increasing the line width to enhance visibility. Additionally, we have replaced the term 'gradient' with 'contour' throughout the figure description.

- Please explain all abbreviations (e.g. HSRL) and reference if appropriate (e.g. for A-Train).

Abbreviations and reference have been checked and added accordingly.

- Lines 104-105: "corresponding to an along-track horizontal resolution of approximately 87 km". Here it should be mentioned that this nominal along-track horizontal resolution of ~87km corresponds to one Basic Repeat Cycle (BRC) also referred as Observation, and pointing to the L2A Algorithm Technical Basis Document (ATBD).

Thank you for the suggestion. We have added this detail and reference into the manuscript following this sentence.

"Each observation by ALADIN integrates laser shots over a 12-second interval, corresponding to an along-track horizontal resolution of approximately 87 km, which is defined as one basic repeat cycle or 'observation', as detailed in the Level 2A Algorithm Theoretical Baseline Document (Flamant et al., 2020a)."

- Line 105: "Each observation is comprised of 24 vertical bins". This is only valid for SCA, the SCAmb used within the study being aligned with only 23 vertical range-bin.

Yes, we agreed on this point. We explained this in the following sentence: "...... the SCAmb method averages extinction values over two consecutive bins."

- Line 121 "Level-2 SCAmb products are used" and line 183 "the ALADIN L2A data from the study period". Here the L2A baseline reference (i.e. 2AXX) should be clearly mentioned as the exact date of downloading from the ESA ADDF.

Baseline reference (baseline 2A11) has been added to both places. Data access information has been added to the **Data Availability** Section.

- Line 174: "official L2A Aeolus processor". The term official could be replaced by operational.

Corrected.

- Line 235, „For the sake of comparison, the ALADIN aerosol retrievals 235 in Fig. 4 (a) have been converted from co-polar to total backscatter coefficients, aligning them with the CALIOP aerosol retrievals in Fig. 4 (b)."I think you mixed up here something. Please check!

Thank you. Checked and corrected.

- Lines 237- 239: Did you use the mean depol value of Calipso or each single profile? At least stating that the depol ratio remains constant with a mean value of 0.32 is quite confusing.

The sentence used to describe the depolarization values in Fig.4(c) is referring to the mean value. We have rephrased the sentence to avoid confusion. "After omitting values below 0 and above 1, the depolarization ratio has an average of 0.32 between altitudes of 2.5 and 7 km."

- Fig. 5. Caption wrong, its extinction not backscatter

Corrected.

- Line 264 "CALIOP's extinction retrieval relies on a predefined lidar ratio tailored for specific aerosol types". Here it might be interesting to point the lidar ratio value assigned to the tropospheric aerosol class highlighted in Figure 8.

The following sentence has been added: "... e.g. 23 ± 5 sr for clean marine, and 44 ± 9 sr for desert dust aerosols at 532 nm".

- 280: „For Fig. 6(b), both measurements show an extinction of ~15 km−1, except where ALADIN observations fail quality-control." How can I see that they fail quality control? Are these the non-existent data points? This is not clear. Please describe better and also which quality control was applied.

We have added a paragraph in Section 2.1 to introduce the QC method used for ALADIN.

"The quality control of ALADIN's Level-2 SCAmb products involves several criteria: the validity of extinction and backscatter coefficient retrievals; the backscatter-to-extinction ratio (BER); Mie and Rayleigh SNRs; estimated errors in extinction and backscatter coefficients; and the accumulated optical depth. These criteria are comprehensively detailed in (Flamant et al., 2020b). ALADIN's L2A processing strategy has a high sensitivity to errors so that small errors in extinction propagate from one bin to the next, often leading to negative extinction coefficients. To mitigate this issue, an additional filtering step is used in this study to eliminate negative extinction coefficients."

- Fig. 6 and 7: Could you also plot the evolution of the Aeolus lidar ratio (after correction).

We are unable to meaningfully plot the evolution of the Aeolus lidar ratio. Aeolus aerosol retrievals are not constrained by lidar ratio, resulting in derived lidar ratios that are often very noisy and require extensive filtering. Additionally, the Aeolus lidar ratio is not directly available from the product. It requires calculation from backscatter and extinction coefficients, each

subject to separate quality control flags. This limitation leads to fewer valid lidar ratios. To address this, we grouped Aeolus measurements over two days and applied filtering to remove abnormal lidar ratios, as depicted by the blue lines in Fig.10.

- Line 293: "This example illuminates a common problem with ALADIN extinction retrieval: retrievals at the base of a thick aerosol layer are very likely to be significantly underestimated or excluded" by quality control due to low SNRs. What does this statement refer to ? Which test cases or analysis have been used to qualify it as a common issue ?

To further support our statement regarding the common issue of ALADIN's low SNRs, we have now referenced two studies (Ehlers et al., 2022; Baars et al., 2020).

- 297: „A noteworthy observation is that ALADIN persistently records an extinction coefficient higher by ~2 compared to CALIOP" I do not see that in you plots.

Thank you for your comment. This is illustrated in Fig.7(e), specifically within the latitude ranges of 8° N to 14° N, and 20° N to 22° N.

- Figure 9. Why do you not provide the Aeolus AOD as well?

In this case of an extreme dust event, a significant number of Aeolus profiles contain bins with missing extinction retrievals. Given Aeolus's 1 km vertical resolution within the dust plume, attempting to integrate the extinction to compute columnar AOD would introduce a substantial bias. Consequently, our focus in this analysis was on utilising Aeolus's lidar ratio to correct CALIOP's extinction and AOD retrievals, rather than on the direct use of Aeolus AOD.

- Fig. 10. Please clearly indicate the wavelength in the plot (355 for Aeolus, 532 nm for CALIOP)

Thank you. Wavelengths have been added to the legend in Fig.10.

- 335: "This scaling method is an approximation, as a different lidar ratio can alter the lidar profile and subsequently affect the retrieval process." Please describe a bit more. I guess you mean the lidar ratio choice already influences the backscatter retrieval during the Klett inversion?

Thank you for the suggestion, and we have revised this sentence: "This scaling method is an approximation, as varying the lidar ratio can influence the lidar profile by impacting the backscatter retrieval during the Klett inversion process. This alteration in backscatter retrieval, in turn, affects the subsequent extinction retrieval."

- Table 1.: I recommend to check the values and complete it with the recent publication of Floutsi et al. (DeLiAn).

Thanks for suggesting this recent reference. Based on the measurements from multiple experiments provided in this reference, we added an extra row of 355 and 532 nm lidar ratio into Table 1.

- Line: 343ff: According to Figure 11, there are no extinction profiles below ca. 1.8 km. Thus I was wondering how did you calculate the total AOD from Calipso? Did you interpolate? Did you just skip the lowermost altitudes? Please clearly describe.

The extinction is not zero below 1.8 km; merely < 1e-2 as shown in Fig.11. The integration of extinction coefficients to obtain AOD has considered all altitudes down to the surface.

- Line 355: "The grouped extinction profile indicate a mean layer AOD 355 of 0.413 between the 0 and 2.4 km layer, accompanied by a considerable standard deviation due to the random distribution of strongly attenuated bins along the satellite track. Conversely, the alternative set of measurements devoid of strongly attenuated bins demonstrates a layer AOD of 1.015 between 0 and 2.4 km." I do not understand this statement at all, please rephrase and describe more!

We have amended the paragraph to enhance the explanation: "……CALIOP measurements with a column AOD below 1.8 often include profiles that feature strong attenuation at the lower boundary of the dust layer, even after applying the described filtering strategy. Specifically, extinction profiles grouped under this threshold demonstrate an average layer AOD of 0.413 for the 0 - 2.4 km layer, with a considerable standard deviation reflecting the presence of strongly attenuated bins. In contrast, profiles with a column AOD of 1.8 or greater, which are free from such attenuation, exhibit a mean layer AOD of 1.015 in the same vertical range. It is this latter set of profiles that tends to yield AOD values consistent with those derived from MODIS observations."

- Conclusion: Please highlight a bit more the synergistic use of Calipso and Aeolus and Seviri for optimum aerosol profiles in this specific dust case.

This sentence has been added to the second paragraph of the conclusion: "This study demonstrates the importance of integrating observations from multiple platforms for optimal aerosol profiling in the context of dust events.".

Language:
- Line 34: „Spaceborne lidars have the advantage of minimal aerosol loading between the instrument and the calibration region." .. I know what you mean with that, but a non-expert user will not understand what is mean there. Please rephrase.

This sentence has been rephrased as: "Spaceborne lidars often self-calibrate by assuming some section of the atmosphere lacks aerosol contamination, typically the stratosphere."

- - Lines 109-110: "Standard Correction Algorithm (SCA)" and "Standard Correction Algorithm middle bin (SCAmb)" should be replaced by "Standard Correct Algorithm (SCA)" and "Standard Correct Algorithm middle bin (SCAmb)"

Thank you. Corrected.

---

## Author Comment (AC4)

*Response to Referee 1*

This manuscript presents the meaningful demonstration of the capabilities of ALADIN in retrieving aerosol optical properties, specifically the backscatter coefficient, extinction coefficient, and lidar ratio. The dust layers' lidar ratios used for CALIOP is also revised according the simultaneous measurements of ALADIN. The manuscript is well written and its contents are of high quality and scientific interest. The benefits of this study would be great for the accurate estimation of Aeolus and CALIOP aerosol data products. Hence, I recommend the acceptance of this manuscript after the necessary revisions.

We greatly appreciate the review and detailed comments provided by the referee. Our responses to the specific comments are as follows.

The specific comments are listed below:

- My main concern about the extinction coefficient/backscatter coefficient comparisons between CALIOP (at 532nm) and ALADIN (at 355nm) is their wavelength dependence. In this manuscript, the authors compare the aerosol products directly without any wavelength convert, even you mentioned it in line 244.

We recognise the limitation in comparing CALIOP (532 nm) and ALADIN (355 nm) due to their different wavelengths. Our study lacks simultaneous data for precise spectral conversion in the specific area and time. While using long-time averaged conversion factors from other experiments is an option, it carries the risk of introducing additional biases. However, as shown in Fig.4 to 7 there are discernible similarities in dust layer retrievals across both wavelengths. Importantly, in the Lidar Ratio and Extinction Retrievals Section, spectral conversion between 355 nm and 532 nm has been applied, allowing for more accurate comparisons of extinction coefficients and AODs at the same wavelength.

- Line 66: could you please give some more detailed comments on why the extinction coefficient is not affected by the misdetection of the cross-polar component?

The retrieval of the extinction coefficient is unaffected by the misdetection of the cross-polar component, as it depends solely on the transmission of light through the atmosphere regardless of polarization. In contrast, backscatter coefficient retrieval, which is dependent on the polarization of scattered light, is indeed influenced by any misdetection of the cross-polar component. For further detail and the relevant equations, please refer to Eq.6.44 for extinction and Eq.6.52 for backscatter in the L2A ATBD.

- Line 104-105: The authors should be aware that the horizontal resolution for Rayleigh channel and Mie channel is different.

Yes, we acknowledge the difference in horizontal resolution between the Rayleigh and Mie channels. To clarify this 87 km resolution is for L2A observation products, we added ".......which is defined as one basic repeat cycle BRC or 'observation'……".

- Line 120: Have the authors ever try to estimate the performance of the products from MLE? You mentioned the MLE method has positive effect on the products retrieve, however, why the Level-2 SCAmb products are applied in your study?

The enhancements of MLE largely arise from the imposition of positivity constraints on optical properties and the employment of a bounded lidar ratio. However, our study's primary objective is to compare the aerosol retrievals of the two lidars with different techniques. Therefore, we use the Aeolus SCAmb products for comparison. This allows us to directly compare the performance and outputs of these distinct lidar technologies. Specifically, the following sentence has been added to the manuscript:

"This approach allows a direct comparison of aerosol retrievals between two different lidar systems, focusing on the performance of the instruments themselves, rather than evaluating advancements in algorithms such as MLE.".

- Figure 1: why the temporal disparity of 9 hours and the maximum spatial difference of 200km are set as thresholds? Is there any physical basis for these selections? For example, wind direction? Air mass transport?

We have revised the collocation paragraph to better explain the selection of specific thresholds. This revision explains the original thresholds used in the collocation database, the narrower thresholds employed in Fig.1 for illustrating the global distribution, and the particular thresholds applied to filter data for the case study.

- The color bar in Figure 1 somehow misleads me. I suggest the authors may use the color bar oppositely, be like Figure 2.

We have now inverted the colourbar in Fig.1 to avoid any confusion and ensure consistency.

- Line 244: the spectral difference between 532 nm and 355 nm could be corrected somehow with the use of typical Angstrom exponent of dust. Have you ever tried to do this work?

We recognise the limitation in comparing CALIOP (532 nm) and ALADIN (355 nm) due to their different wavelengths. Our study lacks simultaneous measurements on Angstrom exponent for precise spectral conversion in the specific area and time. While using long-time averaged Angstrom exponent from other experiments is an option, it carries the risk of introducing additional biases. However, as shown in Fig.4 to 7 there are discernible similarities in dust layer retrievals across both wavelengths. Importantly, in the Lidar Ratio and Extinction Retrievals Section, spectral conversion between 355 nm and 532 nm has been applied, allowing for more accurate comparisons of extinction coefficients and AODs at the same wavelength.

- Line 312: what is the time difference between the measurements from MODIS and CALIOP?

Prior to September 2018, the time difference between MODIS and CALIPSO observations was typically just a few minutes. However, after September 2018 when CALIPSO adjusted its orbit within the A-Train (now referred as C-Train), this time difference increased. In the context of our

case study, the time difference is ~50 minutes. We have included this information in the caption of Fig.8

- The wavelength band that MODIS applied should be pointed out. Hence, we can figure it out whether the wavelength convert should be carried out. From this point of view, the underestimation may be solved.

We have revised the following section in include MODIS AOD wavelength (550 nm), along with the wavelength induced AOD difference at 532 and 550 nm:

"Figure 9 compares MODIS Aqua 550 nm and CALIOP 532 nm AODs for the scene depicted in Fig. 8(a). For this analysis, each CALIOP profile is paired with the nearest valid, cloud-free MODIS Aqua AOD observation. While the typical spectral difference in AODs at 532 nm and 550 nm is ~3-6% (Kim et al., 2013), this difference is relatively small when compared to the larger discrepancies observed within the latitude range of 12° N to 20° N in Fig. 9. Given that CALIOP retrievals have already excluded vertical profiles containing fully attenuated bins, this AOD underestimation cannot be attributed to lost retrievals from the dust's bottom layer."

The technical corrections:
- Line 193: "the blue dots in (d) represent the footprint…" should be changed to "the blue dots in (d) represent the footprints…"

Corrected.

- Please provide the color bars' label for the green/blue gradients in Figure 4 and 5.

Colourbars have been added in Fig.4 (a) & (b), Fig.5 (a) and (b).

- Why there is only one red profile in Figure 4(a) and 5(a) between 12.5 km and 17.5 km? Is it because there is only one measurement case reach that height? Then I would suggest the authors provide the total numbers of measurements at different heights.

The red profile is the averaged value. Additionally, Fig.4(a) & (b), Fig.5(a) and (b) have been updated by adding the total number of measurements at the right margin of each subplot.

- It should be "Comparison of aerosol extinction coefficients…" instead of "Comparison of aerosol backscatter coefficients" in the caption of Figure 5. Please correct it.

Corrected.

---

## Author Response (AR2)

*Response to Referee 1*

My comment on the statement that "ALADIN is more susceptible to signal attenuation from CALIOP" is not appropriately addressed. This statement is not proven in the manuscript and the author's reply is not convincing. The statement cannot be made without support.

We thank the referee once again for their careful review of our manuscript and for the additional comments.

Following your suggestion, we have removed this sentence from the abstract.

*Response to Referee 2*

The authors have addressed all my concerns and the manuscript has been improved. The labels and legends of Figure 3, 6, 7 and 8 are still hard to be read. Please make them larger.

We thank the referee once again for their careful review of our manuscript and for the additional comments.

We have increased the font size in Fig. 3, 6, 7, 8. For the curtain plots in Fig 6 - 8, we implemented a cutoff for the specific latitude range of the plume, and display the VFM data only up to a ceiling altitude of 10km.

*Response to Referee 3*

Thank you for addressing my comments.

I suggest to crop the CALIPSO curtains in Figs. 6 to 8 to cover only those latitudes that are shown in the other panels of the respective figures and to reduce the altitude range to 10 km.

We thank the referee once again for their careful review of our manuscript and for the additional comments.

For the curtain plots in Fig 6 - 8, we implemented a cutoff for the specific latitude range of the plume, and display the VFM data only up to a ceiling altitude of 10km.

*Response to Referee 4*

The authors have done a huge effort and tackled many aspects of the 4 reviewers.

The manuscript is now much clearer and better understandable and has significantly improved. However, it has not yet its full potential. Therefore, I have still some open, minor comments mostly related to responses to my previous concerns.

We thank the referee once again for their careful review of our manuscript and for the additional comments.

• Linear vs. logarithmic plotting

Thanks for making this approach and providing the linear plots to this response.

However, I still believe, as well as reviewer 3, that the linear scale would fit much better to your manuscript especially as you do not focus on the very small backscatter coefficients! Furthermore, it is anyhow questionable how trustable these values in regions with such low aerosol content are as they are beyond the detection threshold anyhow. Thus, please really consider again, which scale you prefer and if it is representative with respect to your argumentation and research focus ( e.g. signal attenuation in strong dust plumes)!

IF you stay with log scale, you should at least plot CALIOP and ALADIN mean profiles in the same plot for comparison, e.g. as Subfigure c.

Yes, we added a direct comparison in linear scale for Figure 5(c).

• Me: In Section 6, important information is missing, e.g., on how the columnar AOD is calculated from Calipso profiles which obviously are not available down to the ground. Currently, the section is really misleading.

Authors: The CALIPSO column AOD is derived by integrating the 532 nm aerosol extinction profile from the 5 km Aerosol Profile Products. Importantly, we have excluded profiles containing fully attenuated bins. This detail has now been added to the last sentence in the first paragraph of Section 4 (updated Section number).

Me again: This is okay, but please state again that you exclude fully attenuated profiles and that in most time the marine boundary layer is not captured and thus the comparison to MODIS is per se biased to underestimation.

We rephrased the two sentences you pointed out in the new comments, to mention the marine boundary layer.

• Me: 297 „A noteworthy observation is that ALADIN persistently records an extinction coefficient higher by ∼2 compared to CALIOP" I do not see that in you plots.
Authors: Thank you for your comment. This is illustrated in Fig.7(e), specifically within the latitude ranges of 8° N to 14° N, and 20° N to 22° N.
Me again: That's the problem of the log scales - it is hardly seen...
Nevertheless, I think the wording "persistently" is not valid in this context, as you also show under estimation ins the same figure. Please rephrase !

We changed "persistently" to "frequently".

• Me: Figure 9. Why do you not provide the Aeolus AOD as well?
Authors: In this case of an extreme dust event, a significant number of Aeolus profiles contain bins with missing extinction retrievals. Given Aeolus's 1 km vertical resolution within the dust plume, attempting to integrate the extinction to compute columnar AOD would introduce a substantial bias. Consequently, our focus in this analysis was on utilising Aeolus's lidar ratio to correct CALIOP's extinction and AOD retrievals, rather than on the direct use of Aeolus AOD.
Me again: This is logical. Correct. But still, It would have be interesting to see the AOD profiles even though comparison results might not be well.

We acknowledge the potential interest in Aeolus AOD values. However, due to QA flags on extinction retrievals and its coarse vertical resolution, in this case study the Aeolus data basically gives the layer-AOD rather total column AOD. To avoid potential confusion in comparing with CALIPSO AOD, we have opted not to include these misleading Aeolus AODs in our analysis.

• Me: Line 343ff According to Figure 11, there are no extinction profiles below ca. 1.8 km. Thus, I was wondering how you did calculate the total AOD from Calipso? Did you interpolate? Did you just skip the lowermost altitudes? Please clearly describe.

Authors: The extinction is not zero below 1.8 km; merely < 1e-2 as shown in Fig.11. The integration of extinction coefficients to obtain AOD has considered all altitudes down to the surface.

Me again: Nevertheless, it is in contradiction with reality as there is a marine BL below. Thus, please make a statement in text. It is still a bit unclear, what conclusion one can draw out of it with these biased extinction profiles.

Yes, we understand neglecting even low extinction values in the MBL (compared with the dust layer above), can result in underestimation of AODs. So as suggested in the new comment, we rephrased the following relevant sentence "It is this latter set of profiles that tends to yield AOD values consistent with those derived from MODIS observations, even though the marine boundary layer is excluded."

• Me: Line 34 „Space-borne lidars have the advantage of minimal aerosol loading between the instrument and the calibration region. " .. I know what you mean with that, but a non-expert user will not understand what is mean there. Please rephrase.

Authors: This sentence has been rephrased as "Space-borne lidars often self-calibrate by assuming some section of the atmosphere lacks aerosol contamination, typically the stratosphere."

Me again: well, this sentence is confusing again and has now a different meaning, what about "Space-borne lidars are often calibrated in atmospheric regions for which a very low aerosol content is assumed, i.e., typically the stratosphere. One further advantage is, that in contrast to ground-based lidars, no significant aerosol contribution is expected between the space-borne lidar and this calibration region."

We have incorporated the suggested revisions for this sentence.

Line 197 of the tracked-changed-highlighted manuscript:
Please state here also the Aeolus Data Version (i.e., 2A11) when mentioning the CALIPSO version.

"(baseline 2A11)" is added.

Figure 5 ff: As a compromise, could you plot the mean of both instruments in linear scale as Sub-Figure c?

We added a direct comparison of two mean profiles in linear scale to Figure 5(c).

Line 378 Please add „...AOD underestimation cannot be attributed to lost retrievals from the dust's bottom layer AND MARINE BOUNDARY LAYER". In general, a bit more discussion on the

underestimation of CALIOP could be made. E.g. not only focussing on lidar ratio improvements but also discuss maybe on missing aerosol layers (marine BL etc.).

We added the potential of losing retrievals for MBL into the sentence.

Line 397 of the tracked-changed-highlighted manuscript: The lidar ratio is actually not derived from CALIOP, maybe „ read out from each individual CALIOP profile" is better?

Agree. We replaced with a word similar to "read" – 'extracted'.

Line 424 of the tracked-changed-highlighted manuscript:
You may add „…It is this latter set of profiles that tends to yield AOD values consistent with those derived from MODIS observations EVEN THOUGH THE MARINE BL IS EXCLUDED"

We added this into the original sentence.